# AAV9:PKP2 improves heart function and survival in a *Pkp2*-deficient mouse model of arrhythmogenic right ventricular cardiomyopathy

## Abstract

**Background** Arrhythmogenic right ventricular cardiomyopathy (ARVC) is a familial cardiac disease associated with ventricular arrhythmias and an increased risk of sudden cardiac death. Currently, there are no approved treatments that address the underlying genetic cause of this disease, representing a significant unmet need. Mutations in *Plakophilin-2 (PKP2)*, encoding a desmosomal protein, account for approximately 40% of ARVC cases and result in reduced gene expression.

**Methods** Our goal is to examine the feasibility and the efficacy of adeno-associated virus 9 (AAV9)-mediated restoration of PKP2 expression in a cardiac specific knock-out mouse model of *Pkp2*.

**Results** We show that a single dose of AAV9:PKP2 gene delivery prevents disease development before the onset of cardiomyopathy and attenuates disease progression after overt cardiomyopathy. Restoration of PKP2 expression leads to a significant extension of lifespan by restoring cellular structures of desmosomes and gap junctions, preventing or halting decline in left ventricular ejection fraction, preventing or reversing dilation of the right ventricle, ameliorating ventricular arrhythmia event frequency and severity, and preventing adverse fibrotic remodeling. RNA sequencing analyses show that restoration of *PKP2* expression leads to highly coordinated and durable correction of *PKP2*-associated transcriptional networks beyond desmosomes, revealing a broad spectrum of biological perturbances behind ARVC disease etiology.

**Conclusions** We identify fundamental mechanisms of PKP2-associated ARVC beyond disruption of desmosome function. The observed PKP2 dose-function relationship indicates that cardiac-selective AAV9:PKP2 gene therapy may be a promising therapeutic approach to treat ARVC patients with *PKP2* mutations.

## Plain language summary

Arrhythmogenic right ventricular cardiomyopathy (ARVC) is a heart disease that leads to abnormal heartbeats and a higher risk of sudden cardiac death. ARVC is often caused by changes in a gene called *PKP2*, that then makes less PKP2 protein. PKP2 protein is important for the normal structure and function of the heart. Human ARVC characteristics can be mimicked in a mouse model missing this gene. Given no therapeutic option, our goal was to test if adding a working copy of *PKP2* gene in the heart of this mouse model, using a technique called gene therapy that can deliver genes to cells, could improve heart function. Here, we show that a single dose of PKP2 gene therapy can improve heart function and heartbeats as well as extend lifespan in mice. PKP2 gene therapy may be a promising approach to treat ARVC patients with *PKP2* mutations.

Arrhythmogenic right ventricular cardiomyopathy (ARVC) is an inherited heart disease characterized by ventricular arrhythmias and progressive cardiac dysfunction[1–5]. Clinical presentation of ARVC progresses from a concealed early phase, a later manifestation of life-threatening ventricular arrhythmias, and ultimately heart failure that requires heart transplant[3,6,7]. ARVC has an estimated prevalence in the general population of 1:1000 to 1:5000 with the mean age of presentation before 40 years old[6,8–11].

Arrhythmic sudden cardiac death (SCD) could be the first symptom, often diagnosed postmortem, of mostly young and athletic patients[7,12].

Mutations in the desmosome gene *Plakophilin-2*, *PKP2*, are the most frequent cause of genetic ARVC and account for approximately 40% of ARVC cases[13–17]. Desmosomes are adhesive intercellular connections that play critical roles in heart development and performance[18,19]. Interactions among desmosome proteins ensure proper structural anchoring and

✉ e-mail: agreer-short@tenayathera.com; jane.yang@tenayathera.com

organization of intermediate filaments, cardiac sarcomere, and other organelles[20,21]. Mutations in the *PKP2* gene are most commonly heterozygous in patients and lead to haploinsufficiency in PKP2 mRNA and protein[22–26]. Reduction of PKP2 protein at the intercalated discs (ID) disrupts desmosomes and other ID structures such as gap junctions (GJs)[6,19,20]. Reduction of Connexin 43 (Cx43), a critical component of GJs, results in compromised electrical coupling and heterogeneous conduction between cardiomyocytes[27,28]. These structural corruptions trigger cell death response, inflammatory infiltration, and metabolic perturbation that underpin clinical manifestations of electrical instability, cardiac structural deterioration, fibrofatty infiltration, and heart failure[24,29–37].

Clinical management of ARVC patients includes lifestyle modification, pharmacological treatment, catheter ablation, implantable cardioverter-defibrillators, and heart transplantation[38,39]. So far, there is no approved treatment that addresses the underlying genetic cause of this disease. It is technically challenging to apply conventional therapeutic approaches to restore defective large cellular structures such as the desmosomes and manage their pleiotropic impact on complex signaling networks. Therefore, a new treatment paradigm[40] that targets the underlying genetic cause of the disease is needed to manage the multiplicity of disease manifestations during disease onset and progression.

In this study, both *PKP2*-deficient human induced pluripotent stem cell-derived cardiomyocytes (iPSC-CMs) and a cardiac-specific knockout, *Pkp2-cKO*, mouse model were utilized to identify the molecular, structural, and functional signatures that recapitulate human ARVC clinical phenotypes. Using the in vivo mouse model, we examined the feasibility and the efficacy of gene replacement by AAV9-mediated exogenous restoration of PKP2. This study demonstrated that ARVC clinical phenotypes, recapitulated by a mouse model, (1) were largely preventable before the onset of cardiomyopathy and (2) can be attenuated after the onset of disease by exogenous restoration of PKP2 expression. Restoration of PKP2 expression led to a highly coordinated and durable correction of structural genes encoding desmosome, sarcomere, and Ca$^{2+}$-handling system, and corrections of multiple signaling pathways of metabolism, inflammation, and apoptosis. Our studies reveal that the desmosome is a fundamental molecular building block in maintaining cellular integrity and functions of cardiomyocytes and the heart. We propose that cardiac AAV9:PKP2 could be a beneficial gene therapy approach to reduce ventricular arrhythmias, slow down adverse right ventricular remodeling, improve heart function, and reduce mortality in ARVC patients with *PKP2* mutations. Correspondingly, the U.S. Food and Drug Administration (FDA) has provided clearance of Investigational New Drug (IND) application to initiate clinical testing of TN-401, Tenaya Therapeutics' AAV9:human PKP2 clinical drug candidate.

## Methods
### Animal studies
Animal studies were performed according to Tenaya Therapeutics' animal use guidelines. The animal protocols were approved by the Institutional Animal Care and Use Committee (IACUC number: 2020.007).

Animals were allowed to acclimate for at least 3 days following shipment. Animals were housed in Innovive racks and disposable cages. The cages are pre-filled with alpha-dri as bedding. Our enrichments have nestlets and Twist-n'Rich for single-housed mice.

Before animal studies were initiated, study designs were created and reviewed that included the objective/hypothesis, treatment dose, sample size, primary outcomes, time points, etc. Baseline (before induction) body weight, age, and gender were used for inclusion and exclusion criteria. Study cages were kept in the same rack through entirety of studies to minimize potential confounders.

For animal studies, ejection fraction was the primary outcome measure used to determine sample size. In addition, the animal sample size per treatment group was determined based on our in-house experience of

conducting efficacy, dose-ranging, and long-term survival benefit studies using AAV9:transgene as test article.

Animals were monitored daily. Humane endpoint was determined based on a body condition score, with weight loss, posture, respiration, activity, and mentation as factors. Animal death was documented and plotted in Kaplan-Meier survival curves and body weight curves. Animals that were euthanized or died prior to the last echocardiography or ECG were excluded from the time progression curve of EF%, RV size, and arrhythmia scores. Animals found dead were excluded from tissue DNA, RNA, or protein analyses. If an animal found dead right after echo or ECG, it would be included in terminal tissue processes.

SOP was strictly followed. The reproducibility was ensured by using animals that were age-matched with evenly distributed body weight and sex. Tamoxifen induction and AAV9 injection were performed by one scientist and echocardiography and ECG were performed by another scientist for consistency. ECG raw traces were examined and double checked by two different scientists. AAV9 was produced and purified following the SOP and each lot titer was determined based on vector genomes. All attempts on replication were successful.

Animals were randomized for either the vehicle control or the AAV9-treated based on body weight and sex that were evenly distributed among treatment groups.

All in vivo animal studies were conducted blinded. Scientists were blinded to group allocation during data collection and analysis.

### Mouse model and route of AAV administration
Tenaya licensed a cardiomyocyte-specific, tamoxifen-activated *Pkp2-cKO* (*αMyHC-Cre-ER(T2)/Pkp2$^{fl/fl}$*) mouse line in the C57BL/6 background from Dr. Mario Delmar, NYU Grossman School of Medicine[31]. The *Pkp2-cKO* animals were induced with tamoxifen at 0.1 mg/g for 3 consecutive days. Tamoxifen injection activates Cre recombinase in Cre-positive animals and induces homozygous deletion of *Pkp2* gene. TN-401 or AAV9:mPkp2 was given as a single dose via retro-orbital injection before or after animals were induced with tamoxifen. *Pkp2$^{fl/fl}$ Cre*-negative littermates were used as wild-type controls. HBSS, as vehicle control, was injected to WT or non-AAV treated *Pkp2-cKO* animals.

WT or *Pkp2-cKO* animals aged 3–10 months were randomized for either the vehicle control or the AAV9-treated based on body weight and gender that were evenly distributed among treatment groups. Additional information on age and gender distribution for each study is available in Supplementary Data 1.

WT male CD1 animals aged about 1 month were used for safety evaluation for TN-401 that was delivered via intravenous injection.

### AAV virus production
AAV production was carried out as previously described[41]. Briefly, HEK293T cells were seeded in a Corning HyperFlask (New York, NY) and triple transfected using a 2:1 PEI:DNA ratio (PEI Max) with a helper plasmid containing adenoviral elements (pHelper), a plasmid containing the Rep2 and Cap genes from the respective AAV serotype, and finally an ITR containing plasmid to be packaged. Three days following transfection cells were harvested and lysed. Virus was purified using iodixanol ultracentrifugation and cleaned and concentrated in Hank's Balanced Salt Solution (HBSS) + 0.014% Tween-20 using a 100-kDa centrifuge column (Amicon, Darmstadt, Germany). AAV was titered using either a PicoGreen or ddPCR assay. AAV particles were diluted to proper titers on the day of animal injection in a formulation buffer, HBSS plus 0.001% Pluronic F68 as a surfactant. The formulation buffer is the carrier buffer for TN-401 or AAV9:mPkp2 to prevent aggregation of capsids. It serves as the vehicle control for the animals 'without treatment' as compared to animals 'with AAV treatment'.

### iPSC-CM culture
iCell Cardiomyocytes2 were thawed according to the manufacturer's instructions (FUJIFILM Cellular Dynamics) and seeded onto Matrigel-

coated plates at a density of 20,000 cells per well of 96-well plates or at a seeding density proportional to the well size. A total of 5 million iPSC-CMs flooded three patterned 4Dcell 35 mm dishes. Seeded cells were maintained in CDI maintenance media (FUJIFILM Cellular Dynamics) for 7–10 days until the day of treatments.

## siRNA knockdown and AAV transduction

To knock down endogenous *PKP2* expression, four independent siRNAs against human *PKP2* (Invitrogen) were tested to confirm silencing at both mRNA and protein level at time points of 2, 4, 6, and 8 days: 4390843 Silencer Select Negative Control No. 1 siRNA; 4392420 Assay Id s531202 Silencer Select Pre-Designed siRNA (#1); 4392420 Assay Id s531203 Silencer Select Pre-Designed siRNA (#2); 4392420 Assay Id s531204 Silencer Select Pre-Designed siRNA (#3); 4392420 Assay Id s10585 Silencer Select Pre-Designed siRNA (#4). RNA sequencing analyses were performed on the above time points using 6-well plates with 4 wells each for the un-transfected, negative control siRNA (siNeg), siRNA #2, #3, or #4 against PKP2 (siPKP2) to select siPKP2 #3 and #4 for functional characterizations of iPSC-CMs in the presence of acute PKP2 silencing. PKP2 mRNA was silenced on day 2 and silencing persistent up to 8 days (the longest time for mRNA monitored by RT-qPCR). PKP2 protein was silenced up to 8 days (the longest time for protein monitored by Western blotting analysis).

A pool of #3 and #4 siRNAs or individual #3 or #4 siRNAs were used to transfect iPSC-CMs at a final concentration of 1.25 or 5 nM using Lipofectamine RNAiMAX (Thermo Fisher Scientific) in CDI maintenance media. Two days after transfection, medium was removed and replaced with fresh CDI maintenance media. Cells were fixed at above time points to determine the effectiveness of silencing using antibody staining and immunofluorescence microscopy. Silencer Select Negative Control siRNA and siRNA #3 or #4 against *PKP2* were applied to iPSC-CMs seeded on three 4Dcell 35 mm dishes on day 7 and fixed on day 10 post silencing for immunofluorescence.

AAV:human PKP2 (AAV:hPKP2), which utilizes an AAV9 variant, CR9-01, and has a higher transduction efficiency to iPSC-CMs than AAV9[42]. This AAV9 variant-based rescue was carried out by AAV transduction at different MOI, multiplicity of infection, on the 3rd day post siRNA-mediated silencing. After overnight transduction of AAV, media was removed the next morning and replaced with fresh CDI maintenance media. TN-401 expression cassette is composed of a codon optimized human PKP2. The codon optimization renders the coding sequence resistant to siRNA-mediated silencing.

## Evaluation of contractility and electrical signals of iPSC-CMs

Contraction of iPSC-CMs was recorded in bright field by SONY SI8000 imaging system and acquired videos were analyzed by DANA Solutions Pulse analysis software (now Curi Bio). Contraction was recorded daily from day 3 to day 8 post AAV transduction and day 5 to day 10 post siRNA treatment. Each siRNA treatment included 6 to 9 wells on two to three independent 96-well plates. Contraction velocity was an average of all wells from the same treatment from the same plate. Averaged numbers of contraction velocity were plotted for each plate from day 3 to day 8 post AAV transduction. Beat period, amplitude, and propagation of electrical signal were detected as extracellular field potential signals from the cardiac monolayers using Axion Biosystems Microelectrode array (MEA) plates[43]. Cell seeding and maintenance on 12, 24, or 96-well plates followed the manufacturer's recommendation. Data were collected at the time points and analyzed the same way as the contraction velocity.

## Immunofluorescence imaging of iPSC-CMs

Cells were fixed in 4% paraformaldehyde for 15 min and permeabilized with PBS + 0.1% Triton-X100 (PBST) at room temperature for 15 min. Cells were washed with PBS three times followed by blocking with PBST + 4% bovine serum albumin (BSA) for 1 h. Cells then were incubated with antibodies against PKP2 (Invitrogen, rabbit polyclonal PA5-53144) at 1:200

dilutions or DSP (Invitrogen, rabbit polyclonal 25318-1-AP; Sigma, mouse monoclonal MABT1492) at 1:100 dilutions or JUP (Sigma, mouse monoclonal P8087) at 1:200 dilutions in PBS + 4% BSA for overnight. After washing with PBS three times, cells were then incubated with donkey anti-mouse or anti-rabbit Alexa Fluor 488, 594 or 647 (Invitrogen) at 1:500 dilutions in PBS + 4% BSA for 2 h. Cells were washed 4 times with DAPI in the final wash, imaged, and acquired on either Leica DMi8 inverted microscope with LAS X3.4.2 Life Science Microscope Software or Molecular Devices ImageXpress Micro Confocal High-Content Imaging System with MetaXpress 6 imaging analysis software. Confocal images were acquired from 9 independent cell areas for each well in a format of 96-well plates. Images in different fluorescence channels were merged using ImageJ software.

## Echocardiography

Transthoracic echocardiography was performed using high resolution micro-imaging systems (Vevo 3100 Systems, Fujifilm VisualSonics) equipped with a 25–55 MHz linear array transducer for mouse heart images acquisition. Briefly, lightly anesthetized spontaneously breathing mice (1–1.5% isoflurane and 98.5–99% $O_2$) were placed in the supine position on a temperature-controlled heating platform to maintain their body temperature at ~37 °C. Parasternal long-axis B-mode tracings of right ventricular outflow track (RVOT) were recorded for RV area measurement, parasternal short-axis M-mode tracings of left ventricle (LV) were recorded for left ventricle internal diameter in end diastole (LVIDd) and LV ejection fraction (EF) calculation.

## Electrocardiogram recordings

Mice were anesthetized with 1–1.5% isoflurane and 98.5–99% $O_2$ via a nose cone (following induction in a chamber containing isoflurane 4–5% in oxygen). Rectal temperature was monitored continuously and maintained at 37–38 °C using a heat pad. Two lead ECG (leads II) were recorded from sterile electrode needles (29-gauge) inserted subcutaneously into the right upper chest with the negative electrode needle and into the left bottom chest with the positive electrode needle. The signal was then acquired and analyzed using a digital acquisition and analysis system (Power Lab; AD Instruments; LabChart 7 Pro software version). ECG parameters were quantified after 1–2 min from the stabilized trace. For spontaneous arrhythmias monitoring, mice were monitored for 30 min after anesthesia induction. The arrhythmia severity scoring system is based on these 30-min recordings, please see Supplementary Table 1 for arrhythmia grade chart. This system has limitations, as the short 30-min recording interval and the anesthesia isoflurane can limit the types of arrhythmias observed. All arrhythmias were spontaneously occurring, with no electrical or chemical stimulation used for induction.

## Mouse histology

Mice were anesthetized with ketamine xylazine cocktail. First, mice were perfused transcardially with PBS. A scalpel was used to cut hearts into two along the coronal plane and drop-fixed overnight in 10% formalin. Fixed heart tissues were sent to Histowiz (https://home.histowiz.com/routine_histology) for trichrome staining and quantification of fibrosis and immunohistochemical staining of PKP2 and Cx43 using rabbit anit-PKP2 and rabbit anti-Cx43 (PA5-53144 and 71-0700).

## Cardiac mRNA and protein analysis

Total RNA was extracted from iPSC-CMs or cardiac RV and LV tissue using the RNeasy Mini or Universal Mini Kit (Qiagen Sciences, 74106 and 73404), cDNA synthesized (Invitrogen SuperScript III First-Strand Synthesis SuperMix for RT) and analyzed by qPCR using Taqman probes to human *PKP2* or mouse *Pkp2* gene, heart failure genes (*Nppa, Nppb*), and fibrosis genes (*Col1a1, Col3a1, and Timp1*). Mouse *Gapdh* served as an internal housekeeping gene control. Absolute transgene mRNA copy number was determined by RT-qPCR using a WPRE-specific RNA standard (GenScript) across six orders of magnitude. WPRE RNA standard sequence and

corresponding primers and probe for RT-qPCR are available in Supplementary Data 1.

Cardiac LV lysates in RIPA lysis buffer were analyzed by immunoblotting with mouse anti-PKP2, rabbit anti-DSP, mouse anti-JUP, mouse anti-Cx43, and mouse anti-GAPDH (SC-393711, 25318-1-AP, P8087, 35-5000-3D8A5, and MA5-15738, respectively). DSP protein level in both supernatant and pellet was analyzed to ensure a complete evaluation.

## Statistics and reproducibility

The numbers of technical and biological replicates and animals for each experiment are indicated in the figure legends. Normality and lognormality tests were performed first to determine whether a dataset is normally distributed. Statistical analyses, ordinary One-Way ANOVA (Tukey's post-hoc test), ordinary Two-Way ANOVA (Tukey's post-hoc test), nonparametric Kruskal-Wallis test with Dunn's correction, and Student's *t* test were performed using GraphPad Prism 9. Significant differences were defined as $p < 0.05$. Error bars in all mouse studies represent SEM (Standard Error of the Mean). Error bars in all cell biology studies represent SD (Standard Deviation). Statistical tests for each individual experiment are provided in the figure legend.

Statistical analyses for RNA sequencing data are detailed below.

## Transcriptional analysis by RNA sequencing

From each replicate, 100 ng total RNA was extracted via the polyA-tail-specific protocol according to Illumina Inc. RNA quality control was performed before library preparation using Agilent TapeStation instrument. The RNA libraries were prepared using a Stranded Total RNA Library Prep with Ribo-Zero Plus kit (Illumina), which also removes ribosomal RNA. The libraries were sequenced as $2 \times 50$ base pair paired-end reads using Illumina NovaSeq 6000 using V1.5 reagent kit on S1 flow cell with an average of 25.66 million reads per each read file (51.32 M reads per sample). After adapter trimming by fastp (version 0.23.3), raw RNA-seq reads from mouse hearts in fastq format were aligned with Salmon (version 1.8.0) to the GENCODE (version M30, July 2022) reference transcript assembly (GRCm39 and Ensembl 107) using best practice parameters to ensure mapping validity and reproducibility (--seqBias --gcBias --posBias --useVBOpt --rangeFactorizationBins 4 --validateMappings --mimicStrictBT2). Next, a script using R package *tximport* was used to generate an expression matrix normalized to transcripts per million (TPM). In this analysis, we only used genes detected in at least 10% of all samples. Protein-coding genes were determined using Ensembl release *mus musculus* annotations (GRCm39, July 2022) and extracted by *biomaRt* (version 2.52.0). Mitochondrial genes were also omitted, followed by renormalization to TPM. These gene expression values were then log2-transformed after addition of 1 as pseudo-count. Expression patterns of key genes associated with functions of interest were visualized across treatment groups with boxplots generated using the *ggplot2* R package. Expression values from both left and right ventricles were included in the boxplots. Relative gene expression levels across groups and two ventricles are also presented in scaled values per gene in the heatmaps. Heatmaps were generated in R using *ComplexHeatmap* package.

For initial assessment and identifying presence of cluster patterns in the transcriptome, Principal Component Analysis (PCA) models were generated in R using the 'prcomp' function from the *stats* package. The first two principal components were used to visualize group level differences across samples in a PCA plot generated using *ggplot2* and *ggfortify* packages with the 'autoplot' function. Differential gene expression analysis was then performed by comparing each two groups of interest using Welch's *t* test on pseudo-log normalized TPM values. The obtained *t* statistics values were used to rank-order the genes for the downstream functional analyses. Volcano plots were then generated to visualize the top positive and negative differentially expressed genes (DEGs) using the *ggplot2* R package. The top DEGs are the set of genes with the highest and lowest t-statistics values. To evaluate functional effects, we performed Gene Set Enrichment Analysis (GSEA)[44] on the gene list pre-ranked by t-statistics obtained from differential gene expression analysis, using the *clusterProfiler* R package. GSEA assesses whether differences in expression of predefined gene sets between two phenotypes are concordant and statistically significant. Gene sets were obtained from positional, curated canonical pathways, transcription factor targets, Gene Ontology, cell type signatures and Hallmark collections in Human MSigDB (v2023.1.Hs)[44–46]. Upon performing GSEA, these gene sets were only considered statistically significant if the false discovery rate (Q value) was less than 0.25 as determined with multiple hypothesis testing correction using the BH-correction method[47]. The normalized enrichment score, which reflects the degree to which a gene set is overrepresented in the ranked list and normalized for gene set size, was used to select significantly altered gene sets. Trends in normalized enrichment scores for some gene sets of interest were shown in heatmaps, which were generated in R using *ComplexHeatmap*.

## Reporting summary

Further information on research design is available in the Nature Portfolio Reporting Summary linked to this article.

## Results

### AAV:PKP2 corrected disease phenotypes in a human iPSC-CM model

To model ARVC disease and identify the molecular, structural, and functional signatures that are fundamental to the disease mechanisms, we carried out RNA sequencing analyses of iPSC-CMs after acute silencing of *PKP2* expression. These studies revealed that the desmosome functions as a signaling hub connecting key structures[48] in cardiomyocytes such that reduction in *PKP2* expression led to down-regulation of structural and functional gene expression encoding components of desmosomes, sarcomeres, intermediate filaments, and ion channels (Fig. 1a). Down-regulation of protein was shown for desmoplakin (DSP), plakoglobin (JUP), myosin-binding protein C3 (MyBPC3), and desmin (DES) (Fig. 1b, the left panel and Supplementary Fig. 1). Trending down-regulation of mRNA was shown for sodium voltage-gated channel α subunit 5 (*SCN5A*) (Fig. 1b, the right panel). PKP2 deficiency resulted in structural disappearance of PKP2 and DSP from the cellular membrane and caused cell disarray of patterned iPSC-CMs (Fig. 1c). In addition, PKP2 deficiency perturbed both contractile (Fig. 1d) and electrophysiological properties of iPSC-CMs (Fig. 1e).

The 1st generation expression cassette was used for iPSC-CM-based studies and the 2nd generation for in vivo mouse efficacy studies (Fig. 2a). Dose-dependent protein expression was evident in iPSC-CMs driven by a cardiac-specific troponin T promoter (Fig. 2b and Supplementary Fig. 2). AAV:human PKP2 (AAV:hPKP2), which utilizes an AAV9 variant, CR9-01, and has a higher transduction efficiency to iPSC-CMs than AAV9[42], restored DSP expression post *PKP2* silencing when compared to the reduced DSP protein without AAV rescue (Fig. 2c, d). AAV:hPKP2 restored contractility as quantified by contraction velocity when compared to the reduced contraction velocity without AAV rescue (Fig. 2e). Using human iPSC-CMs as a cell model for ARVC, AAV:hPKP2 restored desmosomes and rescued contractility in PKP2-deficient iPSC-CMs, suggesting PKP2 governs intrinsic cellular properties of cardiomyocytes.

### *Pkp2-cKO* ARVC mouse model recapitulated the majority of human

**ARVC clinical manifestations.** We used a mouse conditional knockout model to assess the feasibility and the efficacy of AAV9-mediated *PKP2* gene replacement. Consistent with the early observations of this model[31], tamoxifen-induced cardiac deletion of both alleles of *Pkp2* in adult mice did not show overt structural and functional changes at 1 week post induction. Tissue collection at the end of the study and weekly monitoring showed disruption of desmosomes and GJs (Fig. 3a and

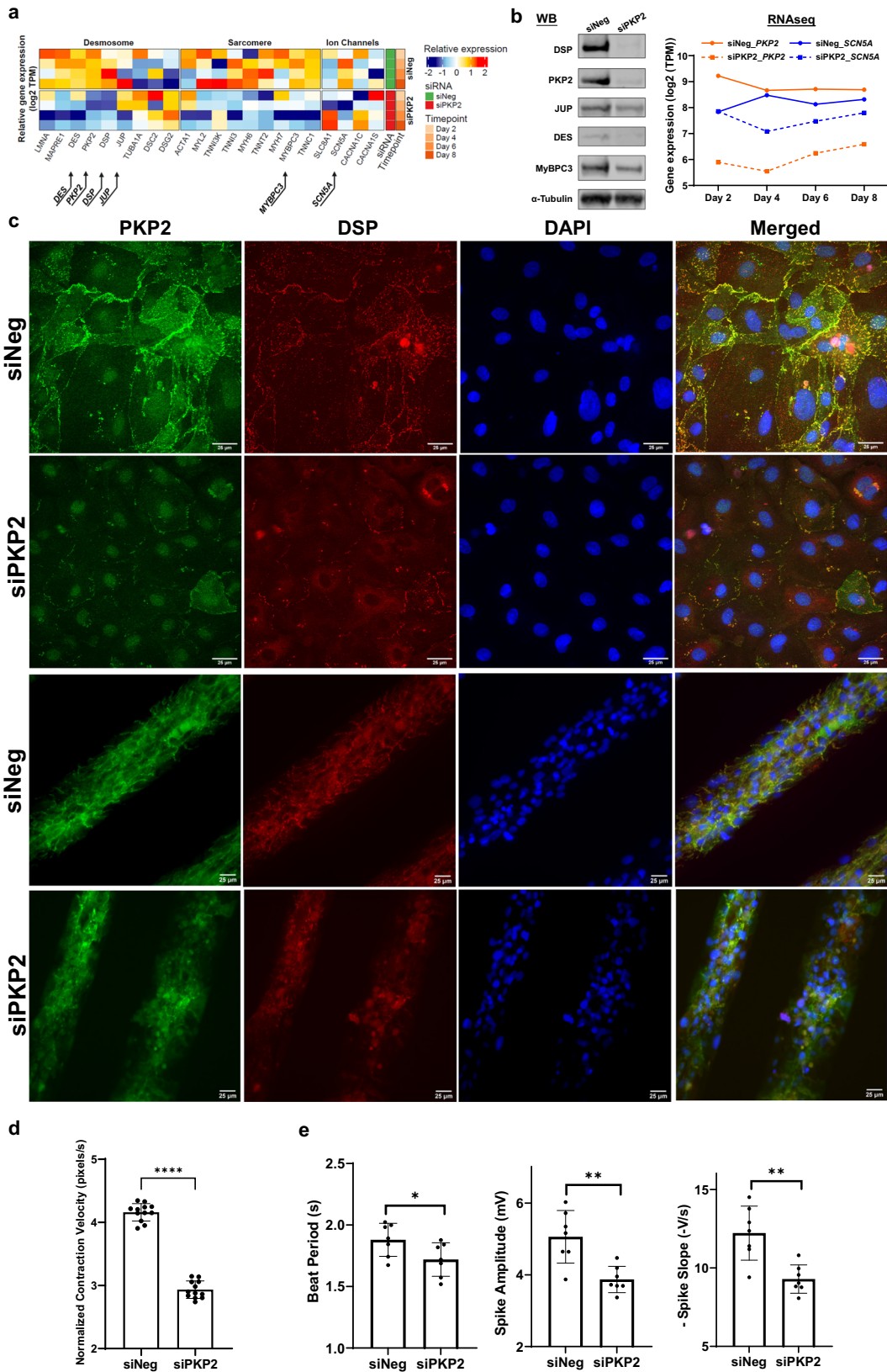

Supplementary Fig. 3), high burden of spontaneous premature ventricular contractions (PVCs) (Fig. 3b) and occurrences of non-sustained ventricular tachycardia (NSVT) (Supplementary Fig. 4), biventricular dilatation (Fig. 3d), and a sharp decline in cardiac function (Fig. 3c) and survival (Fig. 3e) after 3–4 weeks of induced cardiac knock-out of *Pkp2*.

These phenotypes recapitulated human ARVC clinical manifestations. However, unlike in humans, heterozygous disruption of *Pkp2* in mouse hearts did not result in cardiac phenotypes that closely recapitulated human ARVC symptoms[49,50]. Thus, homozygous *Pkp2-cKO* mouse was used as a model of human ARVC.

**Fig. 1 | siRNA-mediated acute PKP2 silencing impacted both cellular structure and functions of human iPSC-CMs. a** Heatmap of RNA sequencing analyses from iPSC-CMs ($n = 1$ for negative control, siNeg, and $n = 3$ biological replicates for siRNAs against PKP2, siPKP2) harvested on day 2, 4, 6, and 8, respectively, after siRNA treatment, highlighting effects on genes encoding components of the desmosome, sarcomere, and ion channels. **b** PKP2 silencing led to reduction in protein expression of DSP, JUP, DES, and MyBPC3 in response to reduced PKP2 protein (Western blot on the left panel, day 8, $n = 2$ biological replicates) and a trending reduction in *SCN5A* mRNA in response to reduced *PKP2* mRNA (RNA sequencing reads from Fig. 1a). **c** PKP2 silencing resulted in disappearance of PKP2 and DSP protein from the cellular membrane (top two rows, day 10, $n = 5$ technical replicates; IXM confocal microscope; 25 μm) and cell disarray in patterned iPSC-CMs (bottom

two rows, day 10, $n = 3$ technical replicates; Leica DMi8 microscope; 25 μm). Immunofluorescent staining: PKP2 in green, DSP in red, and nuclei in blue. **d** PKP2 silencing led to defective contraction as quantified by contraction velocity using Pulse video analysis (day 3 to 8, $n = 12$ technical replicates for each day, $n = 2$ biological replicates) (Curi Bio)[69]. Average nuclear counts from live cells were used to normalize contraction velocity. PKP2 silencing led to (**e**) depressed beat period; depressed amplitude; depressed rate of propagation of electrical signal, detected as extracellular field potential signals from the cardiomyocyte monolayers using Microelectrode array (MEA) plates (day 4 to 10, $n = 18$ technical replicates for each day, $n = 2$ biological replicates) (Axion Biosystems)[43]. Quantified data were presented as mean ± s.d. P value: Student's t test. *$p < 0.05$, **$p < 0.01$, ***$p < 0.001$, ****$p < 0.0001$.

---

**TN-401 or AAV9:mPkp2 treatment largely attenuated disease development and disease progression to mortality in Pkp2-cKO ARVC mouse.** To determine whether the AAV9 expression cassette (Fig. 2a, the 2nd generation) encoding either the human *PKP2* or the mouse ortholog could counteract the effects of cardiac *Pkp2* gene deletion, *Pkp2-cKO* mice were given a single systemic dose via retro-orbital injection of TN-401 (AAV9: human PKP2 at 3E13 vg/kg) or AAV9:mPkp2 (AAV9: mouse Pkp2 at 5E13 vg/kg) 3 weeks prior to tamoxifen induction of cardiac *Pkp2* gene deletion (Fig. 4a). A lower dose level of the human ortholog was selected to limit the risk of over-expressing the human protein in this mouse model. Hank's Balanced Salt Solution (HBSS) was used as the carrier buffer for TN-401 or AAV9:mPkp2 to prevent aggregation of capsids. It was administered as the vehicle control to WT and to *Pkp2-cKO* animals. There were 4 experimental groups: 'WT' and 'cKO' were treated with the vehicle and *Pkp2-cKO* animals treated with either 'TN-401' or 'AAV9:mPkp2' as shown in Fig. 4.

At 4 weeks post gene deletion and 7 weeks post AAV treatment, human or mouse PKP2 inhibited the development of frequent PVCs and the occurrence of NSVT as summarized by a ventricular arrhythmia score (Fig. 4b, Supplementary Fig. 4, and Supplementary Table 1 used as an overall composite score estimating arrhythmia burden), prevented right ventricular remodeling (Fig. 4c), and prevented decline in left ventricular function (Fig. 4d). Frequent PVCs, RV and LV remodeling, and LV function decline were prominent features of *Pkp2-cKO* mice at 4 weeks post gene deletion. TN-401 demonstrated significant efficacy in preventing ARVC development and in extending median lifespan by ≥ 58 weeks, far beyond the 4.7 weeks observed in the vehicle-treated *Pkp2-cKO* animals (Fig. 4e). In this same study, we also evaluated efficacy of AAV9:mPkp2 in *Pkp2-cKO* mice at 3 intervention timepoints and concluded that treatments at 3 weeks before, right after, or 1 week after gene deletion yielded comparable efficacy in EF%, RV remodeling, arrhythmias, and prolonged lifespan of more than 50% of the treated animals by 50 weeks (Supplementary Fig. 5). Overall, these results showed that either the human PKP2 or the mouse ortholog was sufficient to prevent the detrimental cardiac and survival phenotypes of *Pkp2-cKO* mice when delivered in the AAV9 vector. In addition, we were unable to detect sex difference in either *Pkp2-cKO* mice or treatment groups (information on sex distribution in each study is detailed in Supplementary Data 1).

To assess the dose response to TN-401 (Fig. 5) or AAV9:mPkp2 (Supplementary Fig. 6), *Pkp2-cKO* mice were given single systemic treatments via retro-orbital injection of TN-401 at 1E13, 3E13, and 1E14 vg/kg one week after tamoxifen induction of cardiac *Pkp2* gene deletion (Fig. 5a). All animals were sacrificed at 4 weeks post induction (3 weeks post AAV treatment) for histological and expression analyses. TN-401 treatment of *Pkp2-cKO* mice showed dose-dependent efficacy in preventing decline of LV ejection fraction, reducing RV dilation as estimated by RV area normalized to body weight, and a trending reduction in arrhythmias (Fig. 5b). This dose-dependent efficacy was confirmed with larger cohorts of animals in significantly improving LV ejection fraction, reducing RV area and arrhythmia burden, and improving survival (Supplementary Fig. 6).

At molecular level, left ventricle heart tissue showed dose-dependent protein expression of human PKP2 (Fig. 5c, the top panel; Western blot images in Supplementary Fig. 7) as well as corresponding restoration of DSP and JUP, two additional desmosome proteins that were decreased in *Pkp2-cKO* mice (Fig. 5c, the bottom two panels). Connexin 43 (Cx43), a gap junction protein present at intercalated discs, was reduced in *Pkp2-cKO* mice, as shown by immunohistochemistry of heart tissue, and was restored in *Pkp2-cKO* mice treated with TN-401 (Fig. 5d, the top row). TN-401 treatment also significantly reduced fibrosis development and collagen deposition in both right ventricle and left ventricle (Fig. 5d, the bottom row and quantification shown in the right graph). In addition, quantitative analyses of molecular signatures supported that TN-401 treatment reduced mRNA expression of heart failure markers (a significant *Nppa* reduction and a trending *Nppb* reduction), fibrosis, and tissue remodeling genes in the right ventricles (only human PKP2 transgene expression was quantified) (Fig. 5e).

Overall, TN-401 or AAV9:mPkp2 treatment supported a dose-dependent efficacy in improving ARVC phenotypes in *Pkp2-cKO* mouse model of ARVC. TN-401 or AAV9:mPkp2 in the dose-escalation studies demonstrated efficacy at doses ≥3E13 vg/kg in preventing adverse right ventricular remodeling, and improving ventricular function, fibrosis, and electrophysiological properties.

The preventive mode of treatment, dosing before overt structural changes, demonstrated significant benefit of early intervention in largely preventing disease development and extending lifespan. To further examine whether ARVC disease progression could be slowed down or attenuated by restoration of PKP2 expression after overt structural changes, the therapeutic mode of treatment, we dosed animals via retro-orbital injection of AAV9:mPkp2 at 1E14 vg/kg at 2.5 weeks after cardiac deletion of *Pkp2* (Fig. 6a). At 2.5 weeks, overt structural changes were observed that coincided with a rapid development of RV dilation, LVEF decline, and significant ventricular arrhythmias (Fig. 3). Note that the rapid mortality presented by this mouse model (within 3–6 weeks of tamoxifen induction) combined with the relatively slow time course of AAV9 transduction and transgene expression make it challenging to perform the therapeutic mode of treatment. However, at 9 weeks post induction, AAV9:mPkp2 prevented further decline of the left ventricle function when compared to the treated animals at 4 weeks ($p = 0.9416$, ns) (Fig. 6c, f), reduced and reversed right ventricle enlargement when compared to the WT level ($p = 0.6856$, ns) (Fig. 6d, g). Arrhythmia scores showed a trending, but not statistically significant reduction (Fig. 6e, h). This therapeutic mode of treatment reduced mortality throughout one year follow-up with a median lifespan by ≥50 weeks (Fig. 6b), which is comparable to the survival benefit observed in the preventive mode of treatment (Supplementary Fig. 6e).

**Restoration of PKP2 expression led to a highly coordinated and durable correction of PKP2-associated transcriptional networks beyond desmosomes.** It was rather surprising to observe that restoration of a single desmosome component, PKP2, led to significant survival benefits, improved cardiac function, reversed adverse RV remodeling, reduced ventricular arrhythmia frequency and severity, and

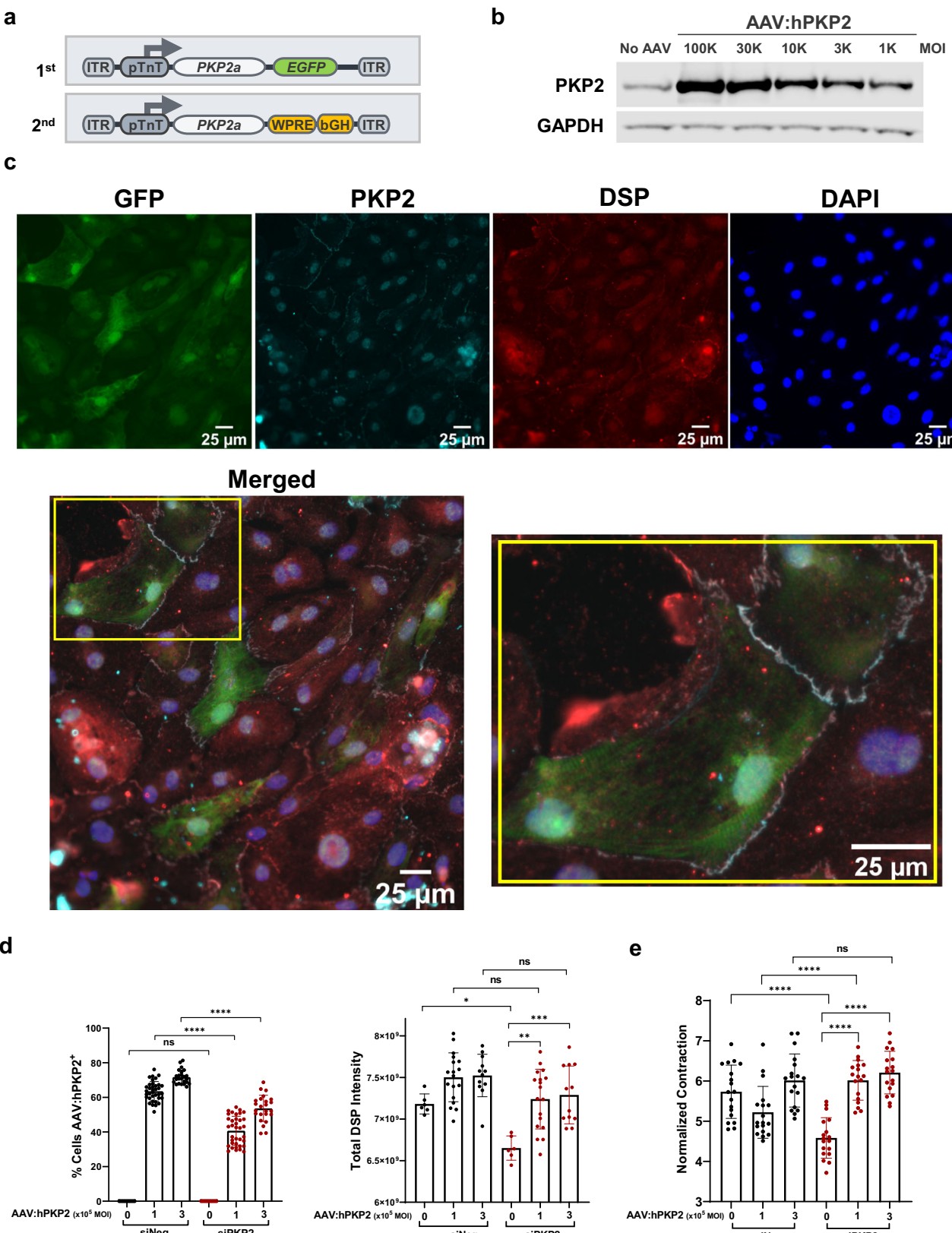

prevented fibrosis. We asked whether "on-target" PKP2 effects possibly extend beyond its effects on the desmosome by evaluating PKP2 dose-dependent response, specifically at the transcriptional level. To our knowledge, there has been no reported study that reveals whether (1) PKP2 dynamically coordinates its gene expression with other desmosome members, and (2) to what extent PKP2 quantitively dictates the state of disease progression. To obtain a deeper understanding, two large-scale RNA sequencing analyses were conducted.

*Pkp2-cKO* mice were given a single systemic dose via retro-orbital injection of TN-401 at 3E13 or 6E13 vg/kg one week before tamoxifen

**Fig. 2 | AAV:hPKP2 transgene restored the expression level of DSP protein and rescued contraction velocity post PKP2 silencing in human iPSC-CMs.**
**a** Schematic representation of the 1st generation and the 2nd generation AAV expression cassette of codon-optimized *PKP2a*. Key 3' elements in AAV expression cassette include Woodchuck hepatitis virus post-transcriptional regulatory element (WPRE), and bovine growth hormone polyadenylation signal (bGH). **b** Western blot analysis showed that the second generation of AAV:hPKP2 is expressed in iPSC-CMs in a dose-dependent fashion by applying viruses at different multiplicity of infection (MOI). **c** At day 10 of PKP2 silencing and day 8 of AAV transduction, GFP expression of the first generation of PKP2 expression cassette was used to label AAV transduced iPSC-CMs. Codon optimization allows the transgene PKP2 resistant to siRNA-mediated silencing. The immunofluorescent mini panels show cells were stained for GFP, PKP2, DSP and nuclei, respectively, with the bottom large panel showing merged channels (Leica DMi8 microscope; 25 µm). Yellow inset was magnified to highlight two GFP cells expressing transgene PKP2 (gray color) at the junction of each other and at the junction of other non-GFP neighbors. **d** At day 10 of

PKP2 silencing and day 8 of AAV transduction, the left bar graph summarized the percentage of cells without GFP (n = 12 technical replicates, n = 2 biological replicates) and with GFP (n = 24–36 technical replicates, n = 2 biological replicates). The right graph showed restored DSP protein expression quantified by total intensity of immunofluorescence signal post PKP2 silencing in the absence (n = 6 technical replicates, n = 2 biological replicates) or the presence (n = 12–18 technical replicates, n = 2 biological replicates) of AAV:hPKP2 transgene. Quantified data were presented as mean ± s.d. Statistical significance was evaluated by ordinary One-Way ANOVA (Tukey's post-hoc test). **e** AAV:hPKP2 showed rescue of contraction velocity post PKP2 silencing in iPSC-CMs (n = 18–27 technical replicates for each day, n = 3 biological replicates). Cell contractility was recorded from day 3 to 8 post AAV transduction and analyzed by Pulse video analysis (Curi Bio). Average nuclear counts from live cells were used to normalize contraction velocity. Quantified data were presented as mean ± s.d. Statistical significance was evaluated by ordinary Two-Way ANOVA (Tukey's post-hoc test). *P* value: \**p* < 0.05, \*\**p* < 0.01, \*\*\**p* < 0.001, \*\*\*\**p* < 0.0001.

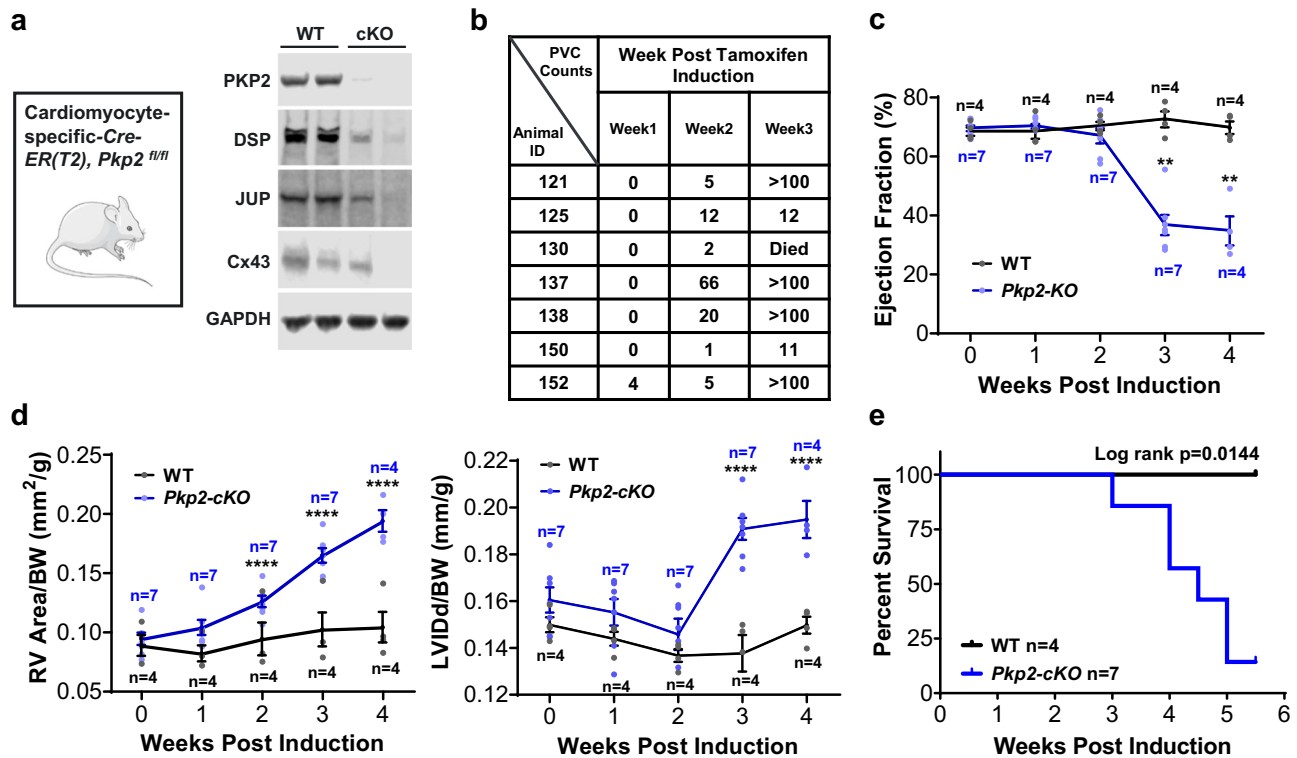

**Fig. 3 | *Pkp2-cKO* ARVC mouse model recapitulated the majority of human ARVC clinical manifestations. a** *Pkp2-cKO* ARVC mice (*αMyHC-Cre-ER(T2), Pkp2^fl/fl*)[31] at ~3 months of age were injected with tamoxifen to induce cardiac knock-out of the *Pkp2* gene. Representative immunoblots showed reduction of desmosome proteins PKP2, DSP, JUP, and GJ protein, Cx43. **b** *Pkp2-cKO* mice developed spontaneous PVCs as observed during 30 min of continuous recording of EKG and reported in the table. Statistical evaluation using nonparametric Kruskal-Wallis test with Dunn's correction showed a significant increase in PVC counts from week 1 to week 3 (*p* = 0.0004). **c** LV performance measured by % ejection fraction sharply declined at 2 weeks post tamoxifen induction. **d** *Pkp2-cKO* mice started to develop

biventricular dilatation between 2 and 3 weeks post tamoxifen induction. RV area (left panel) and LV internal diameter end diastole (LVIDd, right panel) were normalized to body weight. **e** Kaplan-Meier survival curve showed a sharp decline of survival of *Pkp2-cKO* mice beginning 3 weeks post tamoxifen induction. Animals showed symptoms including sudden death, edema, reduced activity, and reduced tolerance to isoflurane beginning 3 weeks post induction. Quantified data were presented as mean ± s.e.m. *P* value: Ordinary Two-Way ANOVA (Tukey's post-hoc test); \*\**p* = 0.002, \*\*\*\**p* < 0.0001 vs. WT at 2, 3, and 4 weeks, respectively. Sample size *n* = 4 and 7 for WT and *Pkp2-cKO*, respectively.

induction of cardiac *Pkp2* gene deletion (Fig. 7a) and cardiac function and arrhythmias were evaluated at 4 and 9 weeks post induction. Mice were sacrificed at 9 weeks post induction and heart tissues were collected for RNA sequencing and quantification of PKP2 RNA and protein expression. At a 2-fold expression difference between 3E13 vg/kg and 6E13 vg/kg doses at 9 weeks (Fig. 7b and Supplementary Fig. 8), we did not observe significant dose-dependent difference in key readouts of EF %, LV mass, RV dilatation, and arrhythmia score, although one out of six

animals at the high dose vs 5 out of nine animals at the low dose had arrhythmia scores ≥1 at 9 weeks post induction (Fig. 7c). We decided to evaluate specific gene classes including desmosome, gap junctions (GJs), sarcomere, ion channels and Ca²⁺ handling systems, heart failure markers, and fibrosis, that have been previously demonstrated to be significant contributors to disease mechanisms (Fig. 7d)[24,29–37]. Comparison between WT vs vehicle treated *Pkp2-cKO* animals showed significant changes in gene expression in these classes and an extensive reversal of

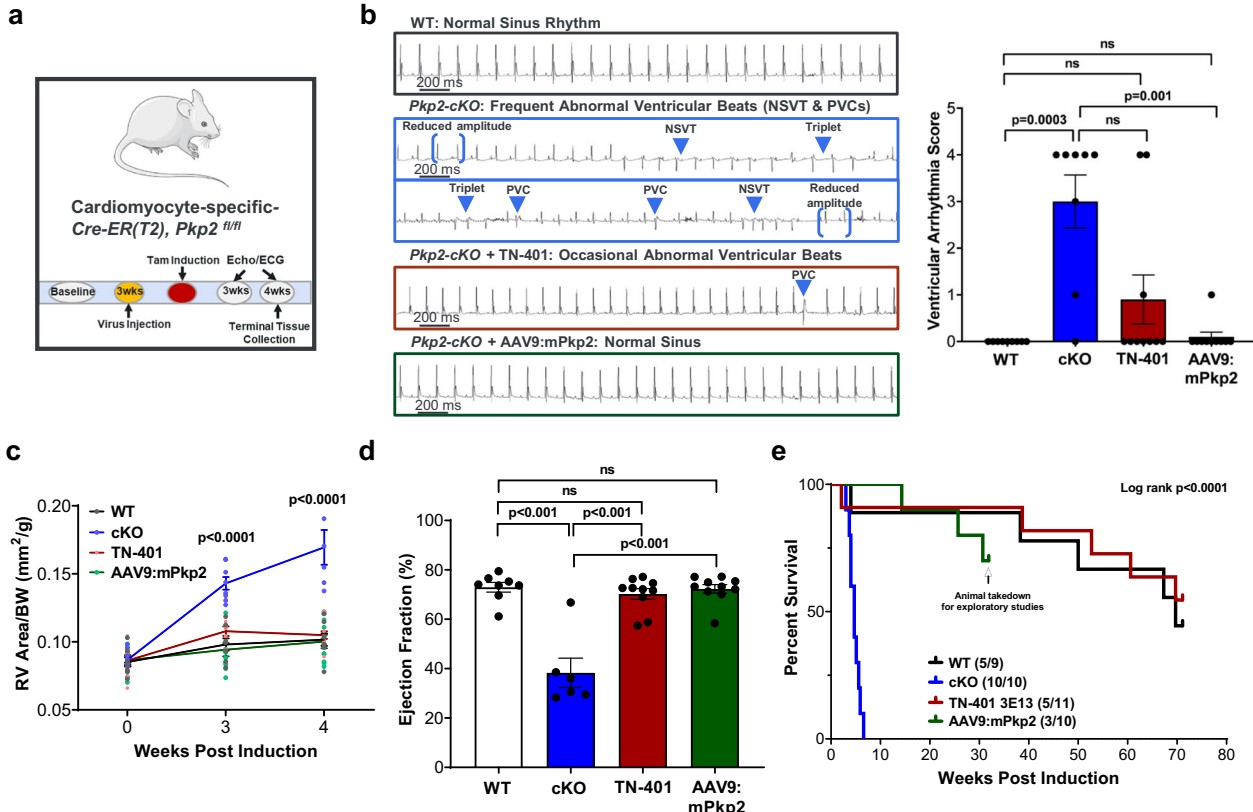

**Fig. 4 | An early and single dose of AAV9:PKP2 significantly reduced arrhythmias, improved cardiac function, and prolonged median lifespan to ≥ 58 weeks post AAV administration. a** Study design to evaluate TN-401 or AAV9:mPkp2 efficacy using *Pkp2-cKO* ARVC mouse model. AAV9 was injected three weeks before gene deletion, TN-401 at 3E13 vector genomes per kilogram bodyweight (vg/kg) and AAV9:mPkp2 at 5E13 vg/kg. Echocardiograph (Echo) and electrocardiogram (EKG) data were collected at week 3 and week 4 post gene deletion. **b** Raw EKG traces showed a significant contrast in spontaneous arrhythmias in *Pkp2-cKO* mice in the absence and the presence of TN-401 or AAV9:mPkp2 treatment. PVCs, premature ventricular contractions; NSVT, non-sustained ventricular tachycardia. The right graph summarized arrhythmia scores representing frequency and severity of ventricular arrhythmias. This arrhythmia score is an overall composite score estimating arrhythmia burden (see Supplementary Table 1). Statistical significance in response to TN-401 or AAV9:mPkp2 treatment was

evaluated using nonparametric Kruskal-Wallis test with Dunn's correction. **c** TN-401 or AAV9:mPkp2 treatment of *Pkp2-cKO* mice showed efficacy in reducing RV dilation as estimated by RV area normalized to body weight (mm²/g) and (**d**) maintaining left ventricular ejection fraction at 4 weeks post gene deletion. Time course of RV dilation was evaluated with ordinary Two-Way ANOVA (Tukey's post-hoc test), statistical significance shown between the vehicle treated cKO animals and TN-401 treated cKO animals. EF% was evaluated with ordinary One-Way ANOVA (Tukey's post-hoc test). **e** Kaplan-Meier survival curve showed that TN-401 extended median lifespan of *Pkp2-cKO* mice by ≥58 weeks post gene deletion. Numbers in paratheses showed dead vs live animals by the time of takedown. Animals treated by AAV9:mPkp2 (in green line) were taken down early for exploratory studies. Quantified data were presented as mean ± s.e.m. Sample size *n* = 9, 10, 11, 10 for WT, cKO, cKO+TN-401, and cKO+AAV9:mPkp2, respectively.

these changes in response to TN-401 (Fig. 7e, genes of interest marked in red). Intriguingly, RNA sequencing analysis at the transcriptional level showed a positive dose correlation to TN-401 among structural genes encoding desmosomes, Cx43, sarcomeres, ion channels and Ca²⁺ handling proteins (Fig. 7f). When examining expression of heart failure markers and fibrosis genes, we noticed a negative dose correlation to TN-401 (Fig. 7f). Therefore, while key functional readouts of efficacy could not be distinguished between dose levels of 3E13 and 6E13 vg/kg, the 2-fold difference in PKP2 transcript levels achieved by these two doses did result in quantitative and dose-dependent changes in transcriptional signatures described above. Based on this observation, we believe that identification of key genes can be informative in associating a transcriptional signature with a particular phase of ARVC disease progression and therefore, may facilitate patient stratification in a more quantitative and precise manner, particularly in early 'concealed' phase when structural changes are not evident.

Transcriptome analyses showed that TN-401 restored expression of structural genes and attenuated expression of genes encoding adverse remodeling factors in a highly coordinated and quantitative fashion. We asked whether such transcriptional response can be sustained to

attenuate disease progression and therefore, extend survival over a longer duration.

As shown earlier in Fig. 6, a single dose of AAV9:mPkp2 treatment at 1E14 vg/kg after overt cardiomyopathy halted disease progression via reversed adverse right ventricular remodeling, improved LV function, a trending reduction in arrhythmias, and extended median lifespan by ≥ 50 weeks post induction of *Pkp2* deletion. Heart tissues collected at 51 weeks post induction of *Pkp2* deletion were analyzed by RNA sequencing (Fig. 8a). Compared to intervention before overt structural changes (the preventive mode, animals dosed at 1E13, 3E13 or 1E14 vg/kg), AAV9:mPkp2 intervention after overt structural change (the therapeutic mode, animals dosed at 1E14 vg/kg) showed comparable efficacy in extending life span at the same dose, 1E14 vg/kg (Fig. 8b). PCA[51] showed that transcriptional profiles of AAV9:mPkp2-treated *Pkp2-cKO* animals were clustered close to WT and distant from vehicle-treated animals, suggesting a normalization of transcriptional landscape close to WT in response to the treatment (Fig. 8c). While the transcriptional profile of low-dose treated animals showed a partial recovery pattern, the transcriptional profiles of the high-dose treated animals effectively overlapped with that of WT samples (Fig. 8c). In addition, when comparing

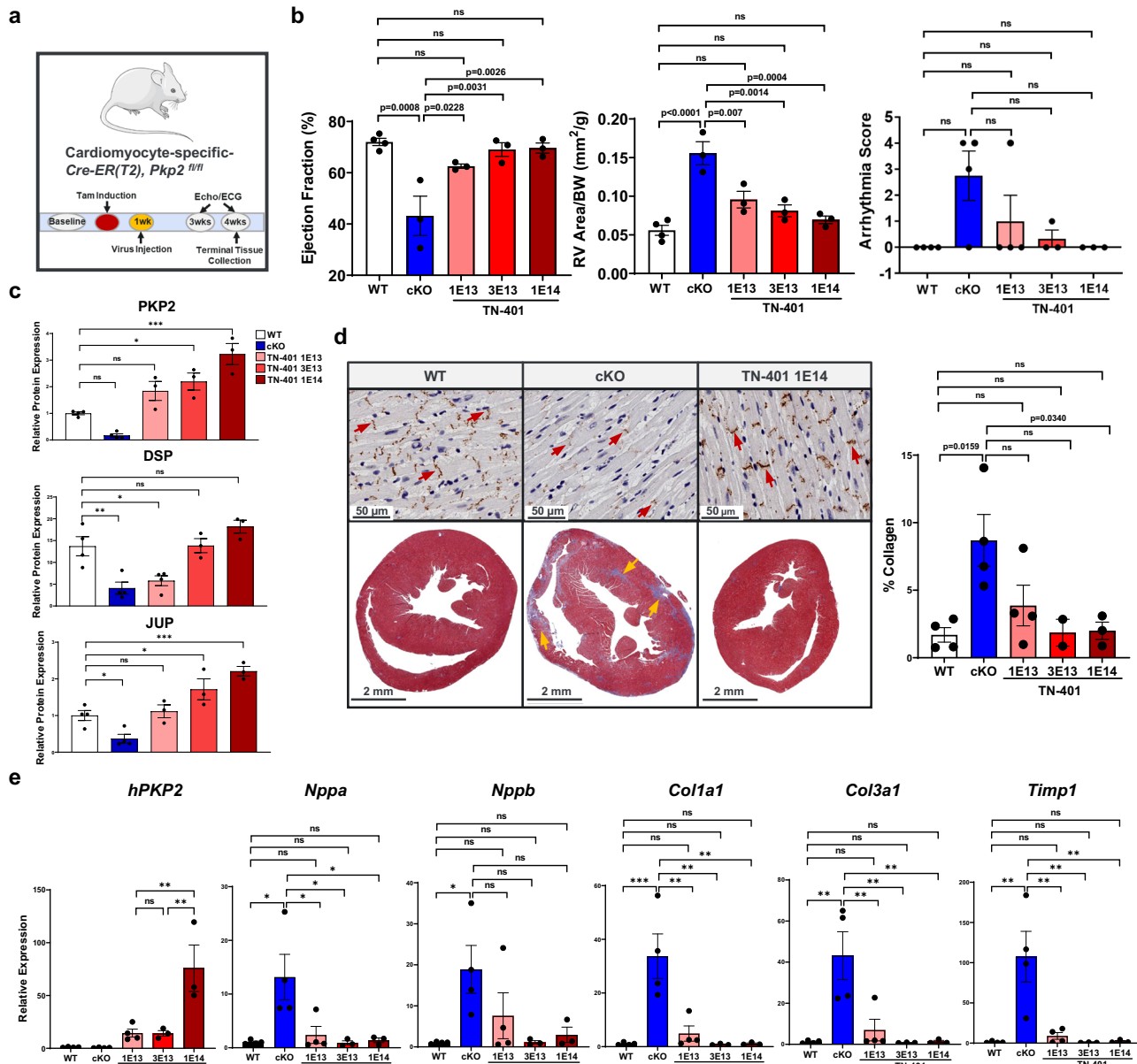

**Fig. 5 | TN-401 dose-dependently reduced arrhythmias, improved heart structure and cardiac function, restored expression of desmosome proteins and Cx43 and prevented development of fibrosis in *Pkp2-cKO* mouse. a** Study design to evaluate dose-dependent efficacy of TN-401 using *Pkp2-cKO* mouse model. Mice were injected with TN-401 at 1E13, 3E13, or 1E14 vg/kg at one week after tamoxifen induction of cardiac *Pkp2* gene deletion. At 4 weeks post tamoxifen induction (3 weeks post TN-401 injection), animals were sacrificed for expression and histological evaluation. **b** TN-401 showed dose-dependent efficacy at 3 weeks in preventing decline of % LV ejection fraction, preventing RV dilation (mm²/g), and a trending improvement in arrhythmia scores. Statistical significance of EF% and RV dilation in response to TN-401 treatment was evaluated with ordinary One-Way ANOVA (Tukey's post-hoc test) and arrhythmia scores with nonparametric Kruskal-Wallis test with Dunn's correction. **c** Semi-quantitative Western blot analyses showed restoration of PKP2, JUP, and DSP protein at 3 weeks post TN-401 treatment. Statistical significance was estimated by ordinary One-Way ANOVA. **d** Immunohistochemistry for the gap junction protein, connexin 43 (Cx43), in heart tissue sections showed restoration of Cx43 expression at intercalated discs (ID) at 3 weeks post TN-401 treatment (the top panels; 50 μm). Red arrows indicate ID. Trichrome staining showed a significant reduction of fibrosis, muscle (red) and fibrosis (blue), in heart sections at 3 weeks post AAV treatment (the bottom panels; 2 mm). Yellow arrows highlight areas with fibrosis in *Pkp2-cKO* mouse heart. The percentage of collagen-positive tissue was quantified and shown in the right graph. Statistical significance was estimated by ordinary One-Way ANOVA (Tukey's post-hoc test). **e** RT-qPCR analyses of RV tissue at 3 weeks post TN-401 treatment showed expression of *hPKP2* transgene and suppression of heart failure markers (*Nppa*) (*Nppb* did not show statistical significance) and fibrosis genes (*Col1a1, Col3a1, Timp1*). *Gapdh* was used as internal control. Statistical significance was estimated by ordinary One-Way ANOVA (Tukey's post-hoc test). Quantified data were presented as mean ± s.e.m. *P* value: *$p < 0.05$, **$p < 0.01$, ***$p < 0.001$, ****$p < 0.0001$. Sample size $n = 4, 4, 4, 3, 3$ for WT, cKO, cKO+TN-401 at 1E13, 3E13, 1E14 vg/kg, respectively. Two animals died between EKG and echocardiogram recordings, and therefore the cKO and cKO+TN-401 at 1E13 vg/kg has $n = 3$ for echocardiogram parameters.

the total number of differentially up- or down-regulated genes relative to the WT animals, the preventive mode and to a lesser degree, the therapeutic mode of intervention showed a significant normalization compared to that between *Pkp2-cKO* and WT animals (Fig. 8d). When comparing vehicle-treated *Pkp2-cKO* animals vs WT, the significant

negatively enriched gene sets identified by Gene Set Enrichment Analysis (GSEA)[44] were mitochondrial dysfunction, cardiac muscle contraction, and cardiac muscle conduction. The top significant positively enriched gene sets were predominantly fibrosis related. Both modes of intervention showed significant reversal of these enriched gene sets with the

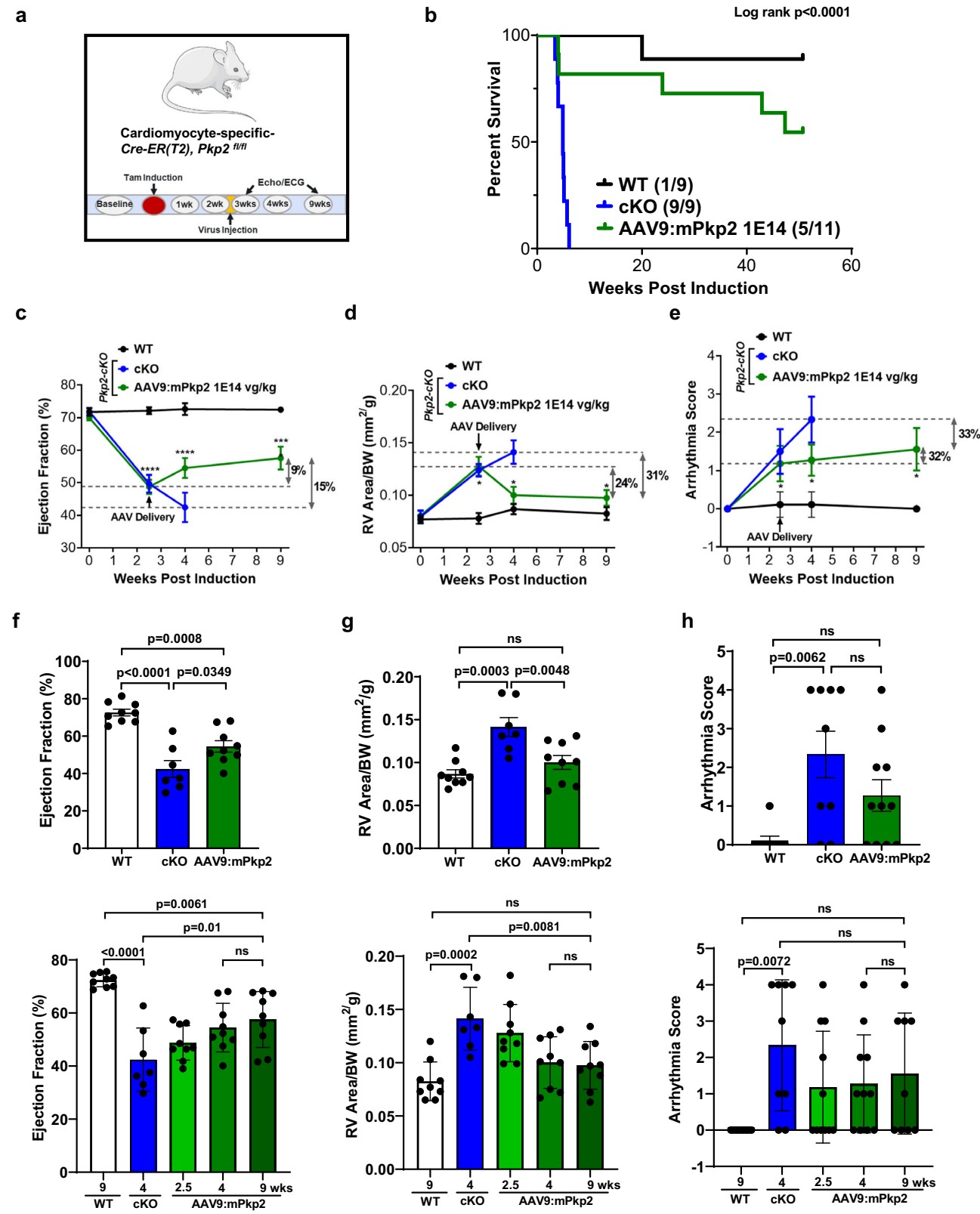

preventive mode supporting the most complete reversal (Fig. 8e). To our surprise, the long-term survival benefit offered by either mode of intervention was supported by a broad spectrum of sustained correction of gene expression encoding components of the desmosomes, sarcomeres, ion channels and calcium handling systems, along with multiple pathways that regulate metabolism, fibrosis, inflammation, and apoptosis as shown (Fig. 8f, g). Once again, both modes of intervention showed significant reversal of these enriched gene sets with the preventive mode effect being most complete (Fig. 8g). Quantitative RT-PCR validated that at the same dose, 1E14 vg/kg, each mode of intervention maintained a similar level of *Pkp2* transgene expression at 51 weeks, suggesting the mode of intervention does not change the durability of the transgene

**Fig. 6 | A single dose of AAV9:mPkp2 after overt cardiomyopathy halted disease progression via reversed adverse right ventricular remodeling, improved LV function, a trending reduction in arrhythmias, and reduced mortality. a** Study design to evaluate AAV9:mPkp2 efficacy using *Pkp2-cKO* mouse model with virus injection at 2.5 weeks after *Pkp2* cardiac gene deletion by tamoxifen induction. **b** Vehicle-treated *Pkp2-cKO* ARVC mice died within 6 weeks of cardiac *Pkp2* gene deletion, in contrast, AAV9:mPkp2 treatment at 1E14 vg/kg significantly reduced mortality and extended median lifespan of *Pkp2-cKO* mice by ≥50 weeks. Numbers in paratheses show dead vs live animals by the time of takedown. AAV9:mPkp2 at 9 weeks post gene deletion and 6.5 weeks post treatment **c, f** prevented further decline of EF%; **d, g** reversed right ventricle enlargement and restored RV size similar to that of WT animals; **e, h** showed a trending reduction in arrhythmias. Statistical significance of EF%, RV area, or arrhythmia scores in time course, **c, d, e**, was evaluated with ordinary Two-Way ANOVA (Tukey's post-hoc test), \*$p < 0.05$, \*\*\*$p < 0.001$, \*\*\*\*$p < 0.0001$, AAV9:mPkp2 treated vs. WT at 2.5, 4, and 9 weeks, respectively. **f–h** the top bar graphs show EF%, RV size, and arrhythmia score at 4 weeks post gene deletion and 1.5 weeks post treatment. The bottom bar graphs show multiple comparisons between treatment groups at different time points. *P* value for all bar graphs in **f–h**: statistical significance of EF% or RV area was evaluated with ordinary One-Way ANOVA (Tukey's post-hoc test) and arrhythmia scores with nonparametric Kruskal-Wallis test with Dunn's correction. All quantified data were presented as mean ± s.e.m. Sample size $n = 9, 9, 11$ for WT, cKO, and cKO+ AAV9:mPkp2 at 1E14 vg/kg, respectively. Four animals died between EKG and echocardiogram recordings at 4 weeks, and therefore cKO and cKO +AAV9:mPkp2 had $n = 7$ and $n = 9$ for echocardiogram, respectively.

expression (Fig. 8h). Expression of fibrosis genes (*Timp1, Col1a1,* and *Col3a1*) were significantly lowered by both modes of treatments at 1E14 vg/kg except for *Col3a1* in left ventricle in response to the therapeutic mode of treatment. Expression of heart failure genes (*Nppa* and *Nppb*) showed trending responses to the preventive mode of treatment relative to the untreated cKO animals (Fig. 8h). In agreement with the observation shown by RNA-seq analyses, fibrosis or heart failure genes were reduced to a lesser extent in therapeutic mode than in the preventive mode among age-matched animals (Fig. 8f, g the top panel).

We concluded that long-term restoration of PKP2 expression by gene replacement approach was correlated with sustained restoration of a broad spectrum of structural genes and pathways, supporting a notion that early intervention is the key to restoring PKP2-associated intrinsic transcriptional networks and their functions and therefore, increasing overall cardiomyocyte fitness to effectively mitigate adverse maladaptive remodeling such as fibrosis as early as possible. These results strongly support that PKP2-associated transcriptional networks can be used to quantitatively evaluate the extent of disease progression and gene therapy efficacy at the molecular level.

**More than 10× an efficacious dose of TN-401 proved to be tolerated in WT CD1 mice.** A six-week pilot tolerability study of TN-401 via intravenous injection at 1E14 or 3E14 vg/kg in WT CD1 mice (Fig. 9a) showed no adverse effects at ≥10x an efficacious dose on body weight (Fig. 9b), heart weight and ventricular functions (Fig. 9c), neutrophil to lymphocyte ratio (Fig. 9d), liver weight and enzyme levels (Fig. 9e), and platelet count and hemoglobin levels (Fig. 9f). Histological analyses showed no TN-401-related changes in heart, lung, liver, pancreas, brain, kidneys, and skeletal muscle examined. Pivotal IND enabling toxicology studies conducted by Tenaya Therapeutics demonstrated safety of TN-401 in both mice and non-human primates. Due to the focus of this report, these data are not included, but were contained in the TN-401 IND application, which has received clearance from the FDA.

## Discussion

Our preclinical results demonstrated that AAV9-based PKP2 gene replacement approach can offer significant survival benefit in repairing cellular structures of desmosome, GJs, and $Ca^{2+}$-handling system, improving cardiac function, reducing PVC frequency and occurrences of NSVT, and preventing adverse fibrotic remodeling in a dose-dependent fashion in a cardiac-specific *Pkp2* knock-out mouse model of ARVC.

About 50% of ARVC patients carry a genetic mutation in desmosome genes and approximately 40% of this patient population carries mutations in *PKP2*. The heterogeneity of clinical symptoms of ARVC indicates genetic composition, maladaptive remodeling, environmental factors, lifestyle, and other unknown factors of individuals likely affect disease onset and progression. Although mutations in desmosome genes have been confirmed as an underlying genetic cause of ARVC by large-scale prevalence studies[13–17], current understanding of the genetic background of this disease is limited and compounded by non-genetic factors and missing molecular links between defective desmosomes and disease development. Using both

iPSC-CM and cardiac-specific *Pkp2-cKO* mouse models of ARVC and AAV9-mediated restoration of PKP2 expression, this study demonstrates that PKP2 is the essential genetic determinant underpinning (1) intrinsic cellular properties of cardiomyocytes; (2) disease onset based on the preventive mode of efficacy studies, which proved AAV9:PKP2 effect to be dose-dependent with significant prevention of multiple disease phenotypes (Figs. 4, 5, 7, and 8); (3) disease progression based on the therapeutic mode of efficacy study, which showed reversal of right ventricle enlargement, halted decline of left ventricle function, a trending reduction in PVC frequency and NSVT occurrence, and most importantly, reduction of mortality (Figs. 6 and 8). As shown in our data, our therapies ameliorated arrhythmia severity by decreasing the PVC burden and the occurrence of NSVT. PVC and NSVT can trigger life-threatening arrhythmias and are associated with increased risk for SCD[52–54]. Due to this associated risk, decreased PVC burden and NSVT frequency can be considered important criteria for the management of ARVC[53].

Heterozygous PKP2 mutations in humans are predominantly truncation mutations resulting from nonsense, frameshift, or splice-site mutations[55–57]. *PKP2* mRNAs containing premature stop codons are subjected to surveillance and degradation by nonsense-mediated decay (NMD) machinery[26,58,59]. Degradation of the mutated mRNA results in haploinsufficiency as shown by reductions of both mRNA and protein in ARVC patient's heart tissues from autopsy, endomyocardial biopsy or explants[22–26]. The preventive efficacy studies demonstrated that restoration of PKP2 expression correlated with restoration of function in a dose-dependent fashion, suggesting that the cellular level of PKP2 precisely and quantitatively dictates a relationship between the cellular input vs the functional outputs under the condition that is minimally influenced by other undefined genetic and nongenetic factors. This precise dose-function correlation of PKP2 (Figs. 5, 7, 8 and Supplementary Fig. 6) possibly addresses the functional consequence of haploinsufficiency in real human cases and further supports the rationale of an AAV-based gene replacement approach. Furthermore, the RNA sequencing analyses revealed a broad spectrum of functional impact by PKP2 deficiency and destruction of desmosomes (Figs. 1, 7, and 8). These results strongly support a gene therapy-based intervention that addresses the root cause and its associated pleiotropism. Mutations in other desmosome genes also lead to destruction of the desmosome, suggesting mechanisms we observed here are likely applied similarly to other desmosome genes in their role of regulating dynamics of desmosome and other multi-unit structures such as GJs, sarcomere, and the $Ca^{2+}$-handling system.

It is significant that restoration of a single component of the desmosome, PKP2, led to significant survival benefits of improved cardiac function, reversed adverse RV remodeling, ameliorated ventricular arrhythmia severity and frequency, and prevented fibrosis. This study demonstrated that "on-target" PKP2 effects extend beyond the desmosomes, at the transcriptional level, to other larger structures or pathways (Figs. 7 and 8). Considering these data, we believe desmosomes dynamically tune and coordinate gene expression of all its components and quantitatively dictate the state and extent of ARVC disease progression as supported by dose dependency of PKP2. Our study showed that

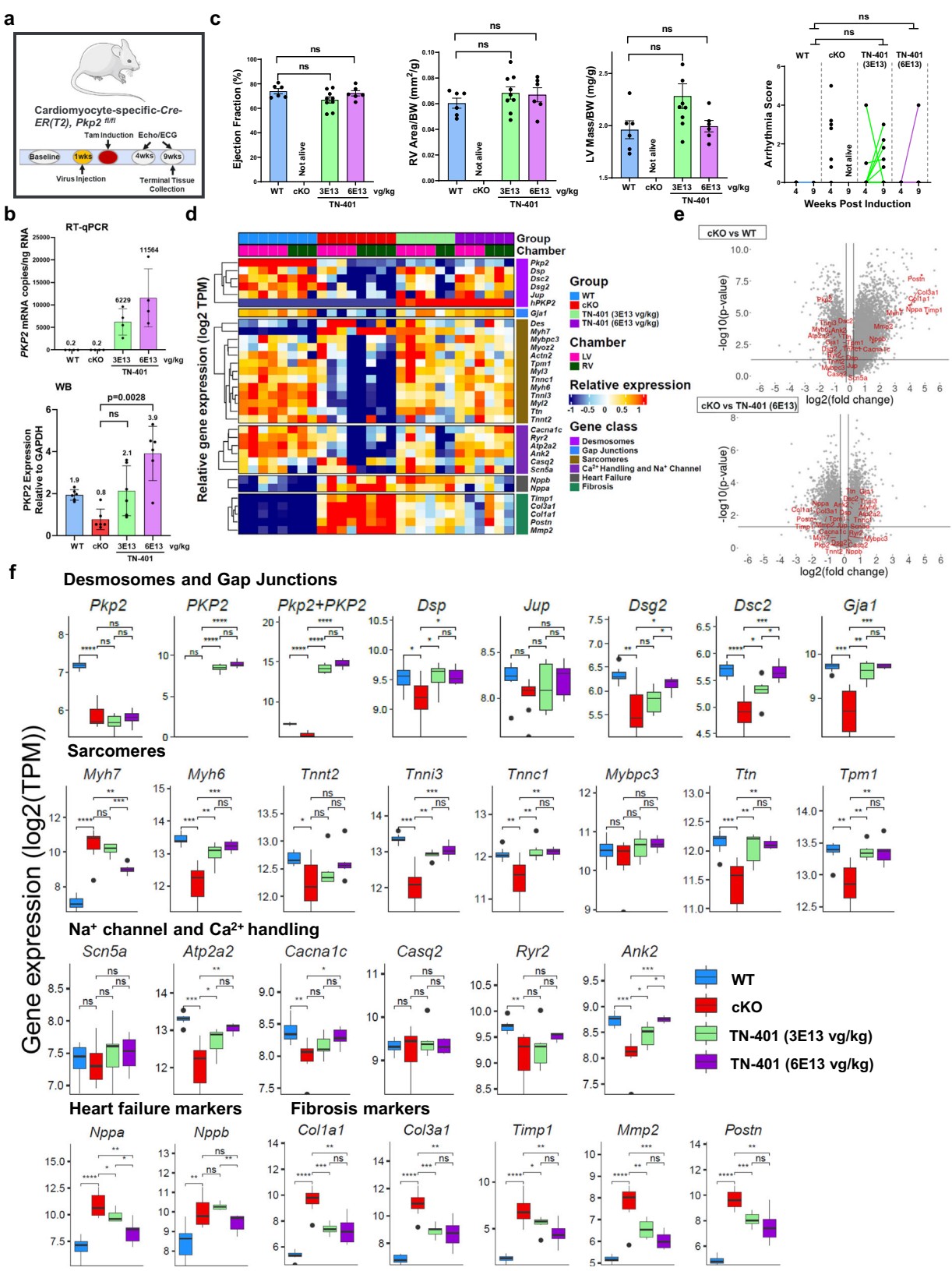

restoration of PKP2 leads to a highly coordinated and durable restoration of structural genes encoding desmosome, sarcomere, and Ca²⁺-handling system components, supporting the desmosome as a fundamental molecular regulatory hub in maintaining cellular integrity and function of cardiomyocytes and the heart, overall (Figs. 7 and 8).

These data further support that an early intervention by gene therapy before overt structural change could project a better prognostic outcome by fixing desmosome-related structural and functional deficits to maintain the overall fitness of cardiomyocytes and to prevent irreversible changes that could compromise the efficacy of gene therapy (Figs. 6 and 8). If overt

**Fig. 7 | TN-401 restored expression of genes encoding desmosome, sarcomere, and Ca²⁺ handling system and attenuated expression of genes encoding adverse remodeling factors in a highly coordinated and quantitative fashion. a** Study design to evaluate TN-401 dose-dependent efficacy at week 4 and 9 post tamoxifen induction of *Pkp2* gene deletion in *Pkp2-cKO* mouse model. Animals were dosed by TN-401 at 3E13 or 6E13 vg/kg at 1 week before induction. **b** Absolute human *PKP2* transgene mRNAs at two doses were quantified in copy numbers per ng of total LV RNA using a mRNA standard (RT-qPCR, the top panel). Endogenous mouse Pkp2 mRNA is not detected by human-specific primers and probes. No statistical comparison was performed between mouse and human gene expression. Human PKP2 transgene protein levels at two doses were compared to the endogenous level of mouse PKP2 protein in WT or *Pkp2-cKO* mouse post cardiac gene deletion by Western blot analysis (WB, the bottom panel). **c** TN-401 treatment at 3E13 or 6E13 vg/kg at week 9 post gene deletion showed comparable efficacy in EF%, RV area (mm²/g, normalized to body weight), LV mass (mg/g, normalized to body weight), and arrhythmia scores. Statistical significance of EF% or RV area was evaluated with ordinary One-Way ANOVA (Tukey's post-hoc test) and 4 to 9-week arrhythmia scores with ordinary Two-Way ANOVA (Tukey's post-hoc test). Quantified data were presented as mean ± s.e.m. **d** Heatmap of gene expression was sorted by treatment groups, heart chambers (LV vs RV), and gene classes. Values were presented in scaled log2-transformed. **e** Volcano plots from differential gene expression (DGE) analysis showed changes between cKO vs WT (WT as reference) (the top graph) and TN-401 at 6E13 vg/kg vs cKO (cKO as reference) (the bottom graph). The X-axis represented log 2 of fold change in gene expression. The Y-axis showed the negative log 10 of $p$ values obtained from DGE analysis for each gene. Genes highlighted in red were selected from each gene class shown in (**d**). **f** Boxplots showed group-wise gene expression for each representative gene of the selected gene classes. Each box showed the distribution of expression values in the following manner: the midline represented the median expression value, the box indicated the interquartile range where the middle 50% of values lie, and the whiskers at the top and bottom of each box represented the range of values outside the interquartile range. The black dots represent values that fall outside the 2nd and 3rd quartiles. Values were log 2 of TPM (Transcripts Per Million) and were aggregated from LV and RV. Comparison $p$ values were calculated by Student's $t$ test: $p$ values: *$p < 0.05$, **$p < 0.01$, ***$p < 0.001$, ****$p < 0.0001$. Sample size $n = 6, 8, 9, 6$ for WT, cKO, cKO+TN-401 at 3E13 and 6E13 vg/kg, respectively.

structural changes have occurred, gene therapy is not expected to be effective, for example, in removing fibrosis or in replenishing more cardiomyocytes. These irreversible changes were investigated in our studies as both preventive and therapeutic modes of intervention showed significant reduction of enriched fibrosis gene sets. However, reduction of fibrosis by the therapeutic mode was to a lesser extent with a concomitant trending higher expression of heart failure genes, *Nppa* and *Nppb* (Fig. 8f, g the top panel and Fig. 8h). The trending higher expression of *Nppa* and *Nppb* in the therapeutic mode of intervention might support observations that the natriuretic peptides encoded by *Nppa* and *Nppb* were functionally implicated in cardiac antifibrotic effects[60]. Significantly, both modes of intervention showed comparable survival benefit, dramatically extending lifespan.

Transcriptome analyses of long-term effects of gene replacement in the therapeutic mode did not have a corresponding age-matched control, which would be the vehicle-treated *Pkp2-cKO* animals that had survived to 51 weeks (Fig. 8). It would be informative to understand the extent of secondary maladaptive remodeling that occurred after overt cardiomyopathy at the age of 51 weeks as the baseline for our therapeutic mode of intervention. This would allow us to determine whether the secondary maladaptive effects responded to the long-term treatment or to evaluate disease regression with pre-existing overt structural changes.

Cardiomyocyte loss due to atrophy or apoptosis puts unique challenges on what might or might not be responsive to therapeutic interventions in adult heart tissue. However, it is intriguing to see proposed reversibility of secondary maladaptive remodeling due to an overall long-term improvement of cardiac performance[61]. The long-term effects by gene replacement targeting cardiomyopathy remain unknown in terms of its potential for disease reversibility. Surprisingly, in our studies, long-term AAV9:PKP2 treatment after overt RV dilatation showed an increased expression of genes underlying sarcomere organization (Fig. 8g, the middle panel with Preventive/WT and Therapeutic/WT). This transcriptional signature could suggest a mechanism of partial regression of adverse structural remodeling due to enhanced sarcomere function in surviving cardiomyocytes. We are currently investigating this possibility.

There are several limitations inherent to this study. First, the majority of PKP2 mutations in human patients are heterozygous germline mutations. The presentation of disease phenotypes in PKP2-mutated ARVC patients progresses gradually with time as discussed in the introduction. In contrast, our proof-of-concept model is a cardiac-specific conditional knock-out model of *Pkp2* with severe phenotypes that develop in a short period of time after gene deletion. Therefore, the robust phenotypes presented in this mouse model could possibly have different sensitivities to gene therapy relative to the more slowly developing phenotypes associated with human disease. Since all proof-of concept studies were conducted in mouse models,

an important caveat is that the clinical efficacy of TN-401 and other PKP2 gene therapy programs or proof-of-concept studies[62–64] (https://rocketpharma.com/aav-presentations-and-posters/; https://www.lexeotx.com/post/lexeo-therapeutics-announces-data-presentations-at-the-26th-american-society-of-gene-cell-therapy-asgct-annual-meeting/) remain to be evaluated in patients with ARVC caused by PKP2 mutations. Second, our study designs and data interpretation were focused on how cardiac function, arrhythmia, and animal survival responded to the gene therapy and on determination of the PKP2 dose-function relationship. We are unable to use this model to evaluate germline impact on organ development and compensatory mechanisms in the presence of PKP2 mutations. Third, although amelioration of arrhythmias in response to the gene therapy was confirmed by EKG and supported by large-scale RNA sequencing analysis on restored gene expression of Ca²⁺ handling system, we do not have direct evidence to support a functional restoration of Ca²⁺ handling at the cellular level. We are currently investigating Ca²⁺ handling using isolated primary cardiomyocytes from heart tissue in response to the gene therapy.

In the studies we present, *Pkp2-cKO* animals treated with the formulation buffer served as the vehicle control to the animals treated with TN-401 or AAV9:mPkp2. Although buffer vehicles, empty capsids or AAV expressing GFP served as controls in other gene therapy proof-of-concept studies[65–67], each type of control has its pros and cons regarding, for example, virus production, animal cost, or cardiac toxicity[68]. Buffer vehicles cannot anticipate the capsid-directed effect or any effect due to overexpression of gene-of-interest to host cells. However, the capsid-directed effect or any effect due to overexpression of PKP2 gene were comprehensively evaluated in pivotal IND-enabling toxicology studies conducted in both wild-type mice and non-human primates by Tenaya Therapeutics. Due to the focus of this report, these data are not included, but were contained in the TN-401 IND application, which has received clearance from the FDA (https://investors.tenayatherapeutics.com/news-releases/news-release-details/tenaya-therapeutics-announces-fda-clearance-begin-clinical).

We are actively developing methods that allow us to determine the percentage of cardiomyocytes versus non-cardiomyocytes transduced by either TN-401 or AAV9:mPkp2. Therefore, we acknowledge that we are currently unable to determine whether desmosomes are formed between cardiomyocytes and non-cardiomyocytes and whether non-cardiomyocytes could possibly contribute to ARVC disease mechanisms.

We used surface EKGs to monitor animal arrhythmias, and we acknowledge the limitation of using this system as animals must be anesthetized. This allows for only short recordings (we chose 30 min) and may also attenuate arrhythmia severity. We were able to observe a high frequency of PVCs and NSVT, but not life-threatening arrhythmias such as sustained VT or ventricular fibrillation. Additionally, we were unable to assess animal death modality.

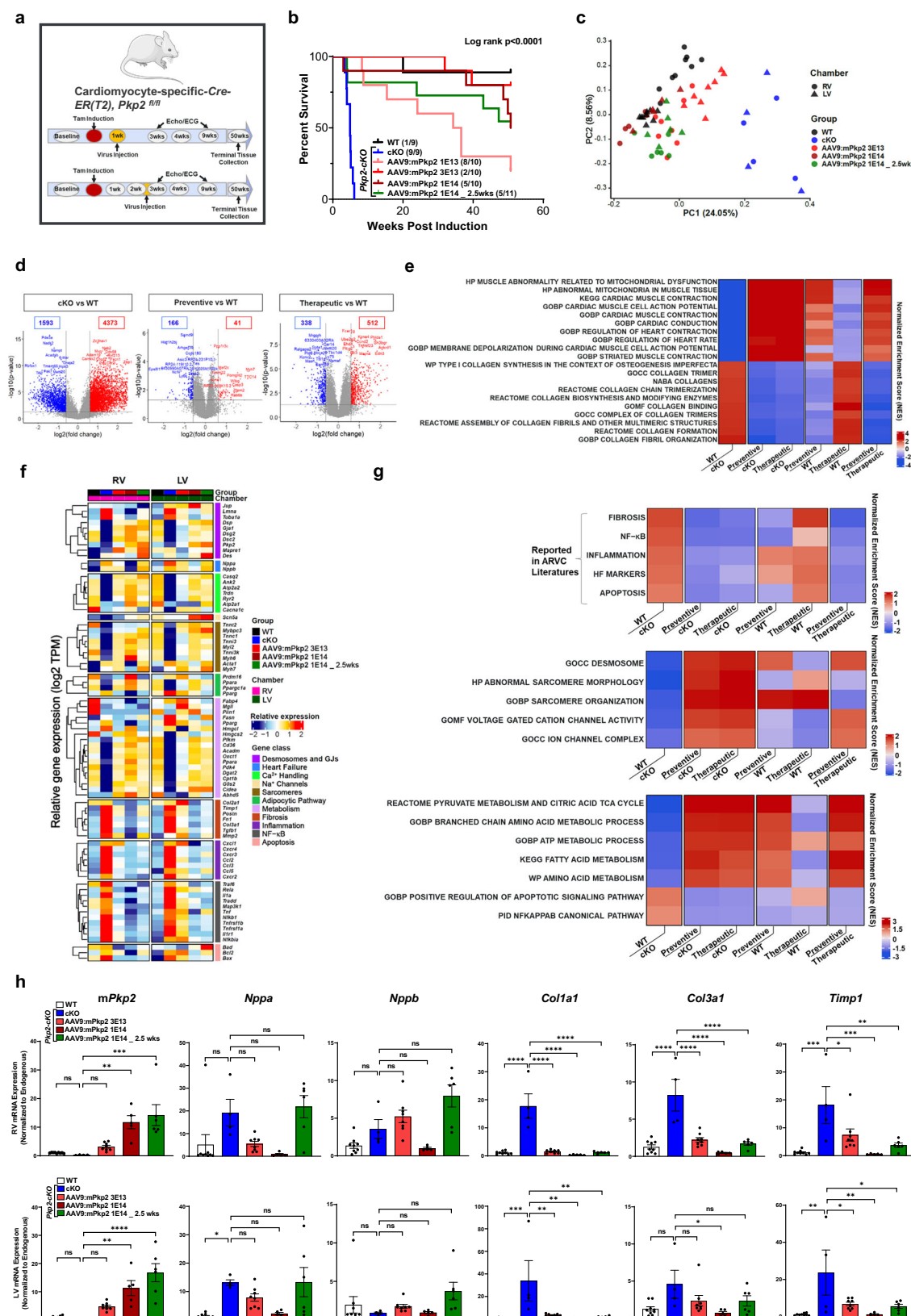

## Conclusions

Using a cardiac KO mouse model of ARVC, we identified fundamental mechanisms behind disruption of PKP2-associated desmosome function and revealed its broad impact at the transcriptional level on GJs, sarcomere, ion channels and Ca²⁺ handling systems, and multiple pathways that critically regulate metabolism, inflammation, apoptosis, and fibrosis. These studies highlight the importance of identifying key transcripts in characterizing ARVC disease state and disease progression in a more quantitative and precise manner. Our efficacy data with PKP2 gene therapy and mechanistic analyses shed more light on understanding ARVC etiology and

**Fig. 8 | Long-term durable expression of AAV9:mPkp2 significantly reduced mortality after disease onset and sustained a broad spectrum of pathways that were perturbed in *Pkp2-cKO* ARVC mouse and restored by the gene therapy.**
**a** Study design to evaluate AAV9:mPkp2 efficacy in reducing mortality at 51 weeks post tamoxifen induction of *Pkp2* deletion in *Pkp2-cKO* mouse model. AAV9:mPkp2 was dosed at 1E13, 3E13, and 1E14 vg/kg either 1 week before the induction (the preventive mode of treatment) or at 1E14 vg/kg at 2.5 weeks after induction (the therapeutic mode of treatment). **b** Kaplan-Meier curve showed percent survival for each mode of treatment for 51 weeks post *Pkp2* deletion. Numbers in paratheses show dead vs live animals by the time of takedown. **c** Principal Component Analysis (PCA) showed clusters of gene transcripts from WT (animals taken down at 51 weeks post induction), vehicle treated *Pkp2-cKO* animals (animals taken down at 4 weeks post induction), and AAV9:mPkp2 treated animals (animals taken down at 51 weeks post induction). Principal components 1 and 2 were visualized in X and Y axes. Numbers in paratheses represented variation in the data explained by each PC. **d** Volcano plots from differential gene expression analysis showed changes between cKO vs WT, preventive vs WT, and therapeutic vs WT. Numbers in boxes represented down-regulated and up-regulated genes in blue and red, respectively. The X-axis represented log 2 of fold change in gene expression. The Y-axis showed the negative log 10 of *p* values obtained from DGE analysis for each gene. **e** Top 10 positively and top 10 negatively enriched cardiac gene sets were shown with FDR Q value less than 0.25 in Gene Set Enrichment Analysis (GSEA) of *Pkp2-cKO* vs WT (the far-left column). These enriched gene sets were used to

compare preventive vs *Pkp2-cKO*, therapeutic vs *Pkp2-cKO*, preventive vs WT, therapeutic vs WT, and preventive vs therapeutic. **f** Relative gene expression of selected genes was measured by RNA-seq. Samples were sorted by treatment groups and RV and LV chambers. Genes were categorized by gene classes. Each column depicted a scaled average across samples of each treatment group. Number of animals in each treatment group used for RNA sequencing were 9 WT, 4 cKO, 2 at 1E13 vg/kg (not included on the Heatmap), 8 at 3E13 vg/kg, 5 at 1E14 vg/kg, and 6 at 1E14 vg/kg (the therapeutic mode) with both RV and LV collected. **g** GSEA heatmap presented positively and negatively enriched cardiac gene sets in *Pkp2-cKO* vs WT. These enriched gene sets were used to compare preventive vs *Pkp2-cKO*, therapeutic vs *Pkp2-cKO*, preventive vs WT, therapeutic vs WT, and preventive vs therapeutic. The top heatmap showed known gene sets reported in ARVC literatures (Fig. 7d); middle and bottom heatmaps showed annotated gene sets of Canonical Pathways and Gene Ontology groups from Human MSigDB database (v2023.1.Hs) and had Q value < 0.25 (See Methods). **h** RT-qPCR analyses showed expression of a total of mouse *Pkp2* mRNA (including mouse transgene mRNA), heart failure marker genes, *Nppa*, *Nppb*, and fibrosis genes, *Timp1*, *Col1a1*, and *Col3a1*, in RV (top row) and LV (bottom row) at 51 weeks post *Pkp2* deletion. Statistical evaluation was performed using ordinary One-Way ANOVA (Tukey's post-hoc test); *P* values: *$p < 0.05$, **$p < 0.01$, ***$p < 0.001$, ****$p < 0.0001$. Sample size $n = 9, 9, 10, 10, 10, 11$ for WT, cKO, cKO+AAV9:mPkp2 at 1E13, 3E13, and 1E14 vg/kg 1 week post cKO induction, cKO+AAV9:mPkp2 at 1E14 vg/kg 2.5 weeks post cKO induction, respectively.

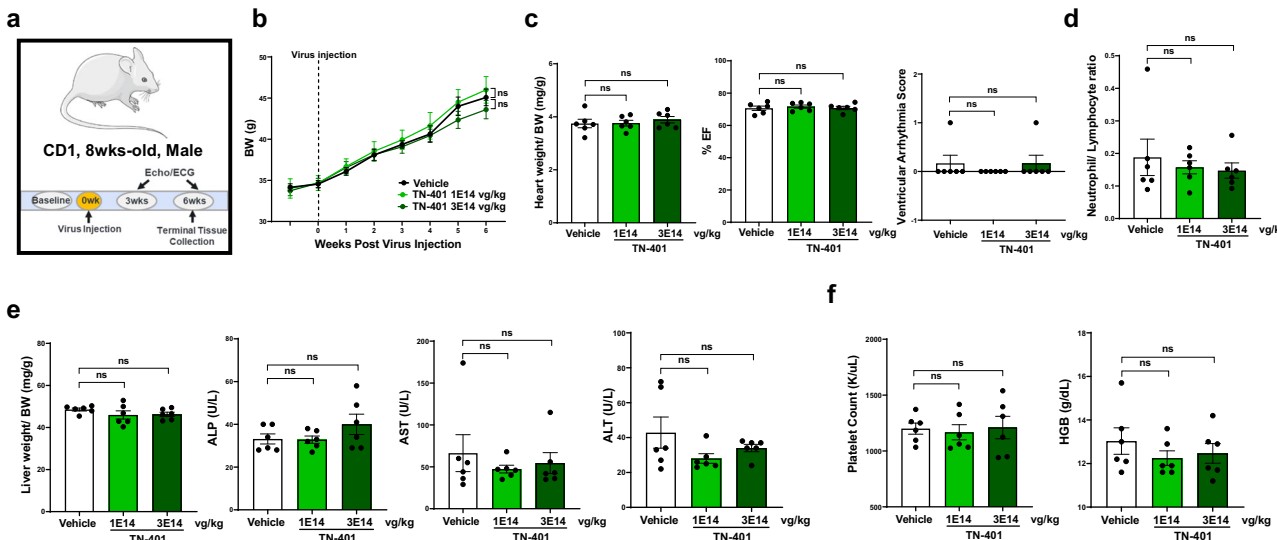

**Fig. 9 | TN-401 pilot safety study in WT mouse proved to be tolerated at ≤ 10X an efficacious dose. a** Study design to evaluate TN-401 tolerability in WT CD1 mice. Mice were injected with TN-401 at 1E14 and 3E14 vg/kg, respectively, after baseline readings of body weight, echocardiography, and EKG. Readings post virus injection were recorded at 3 and 6 weeks, respectively, including echocardiography and 30-min ECG. Mice were sacrificed in week 6 and tissues and blood samples were collected. **b** Body weight progression for 6 weeks. **c** Heart weight normalized to body weight, %EF, and ventricular arrhythmia score at 6 weeks. **d** Neutrophil to lymphocyte ratio at 6 weeks. **e** Liver weight normalized to body weight and live function

tests, alkaline phosphatase (ALP), aspartate transaminase (AST), alanine transaminase (ALT) at 6 weeks. **f** Platelet counts and hemoglobin (HGB) amount at 6 weeks. *P* value: statistical significance of BW progression at 6 weeks was evaluated with ordinary Two-Way ANOVA; heart weight/ bodyweight, EF%, neutrophil to lymphocytes ratio, liver weight/body weight, ALP, AST, ALT, platelet counts, or HGB were evaluated with ordinary One-Way ANOVA (Tukey's post-hoc test); and arrhythmia scores with nonparametric Kruskal-Wallis test with Dunn's correction. All quantified data were presented as mean ± s.e.m. Sample size $n = 6$ for all groups.

in practice, may help patients and physicians to make decisions regarding disease management and treatment.

## Data availability
All data generated or analyzed during this study are included in this article as Source data. Source data for Figs. 1–9 and Supplementary Figs. 5–6 can be found in Supplementary Data 1. The list of mouse age and gender and sequences for RT-qPCR are available in Supplementary Data 1. RNA sequencing datasets are available at GSE253226. All other data are available from the corresponding author (or other sources, as applicable) on reasonable request. Queries related to arrhythmias should be addressed to

Amara Greer-Short. Any other queries should be addressed to Zhihong Jane Yang.

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

## Acknowledgements
We thank Dr. Mario Delmar and NYU for licensing *Pkp2-cKO* mouse.

## Author contributions
Conceptualization: Z.J.Y. and J.Y.; Methodology: I.W., A.Z., A.G.-S., J.A.A., A.E.T., M.VP., E.X., C.R., N.R., B.L., T.W.C., J.W., A.S., S.J., C.D.-H., C.G.G., J.M., K.R., Y.H., R.B., S.S., J.H., J.R.P., J.Y. and Z.J.Y.; Formal analysis: I.W., A.Z., A.G.-S., J.A.A., A.E.T., J.Y. and Z.J.Y.; RNA sequencing analysis: R.S. and F.F.; Investigation: Z.J.Y., J.Y., A.Z., F.F. K.N.I. and T.H.; Data curation: I.W., A.Z., A.G.-S., J.A.A., A.E.T., R.S., F.F., J.Y. and Z.J.Y.; Writing-original draft preparation: Z.J.Y.; Writing, Review, and Editing: Z.J.Y., A.G.-S., R.S., F.F., J.A.A., A.E.T., K.N.I., T.H. and A.Z.; Supervision: Z.J.Y., J.Y., A.G.-S., F.F., S.J. J.R.P., X.S., F.J., K.G., K.N.I. and T.H.

## Competing interests
The authors declare no competing interests.

## Additional information

**Iris Wu** [1,2,3], **Aliya Zeng** [1,3], **Amara Greer-Short** [1] ✉, **J. Alex Aycinena** [1], **Anley E. Tefera** [1], **Reva Shenwai** [1], **Farshad Farshidfar** [1],
**Melissa Van Pell** [1], **Emma Xu** [1], **Chris Reid** [1], **Neshel Rodriguez** [1], **Beatriz Lim** [1], **Tae Won Chung** [1], **Joseph Woods** [1], **Aquilla Scott** [1],
**Samantha Jones** [1], **Cristina Dee-Hoskins** [1], **Carolina G. Gutierrez** [1], **Jessie Madariaga** [1], **Kevin Robinson** [1], **Yolanda Hatter** [1],
**Renee Butler** [1], **Stephanie Steltzer** [1], **Jaclyn Ho** [1], **James R. Priest** [1], **Xiaomei Song** [1], **Frank Jing** [1], **Kristina Green** [1],
**Kathryn N. Ivey** [1], **Timothy Hoey** [1], **Jin Yang** [1,4] & **Zhihong Jane Yang** [1,4] ✉

[1]Tenaya Therapeutics, South San Francisco, CA 94080, USA. [2]Present address: University of Michigan, Department of Molecular and Integrative Physiology, Ann Arbor, MI 48109-5622, USA. [3]These authors contributed equally: Iris Wu, Aliya Zeng. [4]These authors jointly supervised this work: Jin Yang, Zhihong Jane Yang.
✉e-mail: agreer-short@tenayathera.com; jane.yang@tenayathera.com

