## [Peer Review File · Communications Medicine]

This manuscript has been previously reviewed at another Nature Portfolio journal. This document only contains reviewer comments and rebuttal letters for versions considered at Communications Medicine

REVIEWERS' COMMENTS:

Reviewer #1 (Remarks to the Author):

The Authors present a preclinical study that tests a gene replacement strategy for treating arrhythmogenic cardiomyopathy caused by loss of function mutations in the cardiac Plakophilin gene (PKP2).

The investigation is performed in a PKP2 KO murine model. This model presents the limitation that, at variance with humans, the mice require the ablation of the protein to approximate the phenotypical traits seen in the heterozygous state in Humans. To address the limitation of the murine model, the authors also perform experiments in human induced pluripotent stem cell-derived cardiomyocytes.

The authors conclude that their studies demonstrate that “Restoration of PKP2 expression led to a highly coordinated and durable correction of structural genes encoding desmosome, sarcomere, and Ca²⁺-handling system, and corrections of multiple signaling pathways of metabolism, inflammation, and apoptosis.

Detailed comments to the authors

Lines 89-92 The authors state that (they) propose that cardiac AAV9:PKP2 could be a beneficial gene therapy approach to reduce ventricular arrhythmias, slow down adverse right ventricular remodeling, improve heart function and reduce mortality in ARVC patients with PKP2 mutations. This hypothesis will be tested in a First in Human clinical study as the authors state that FDA has provided clearance of an Investigational New Drug (IND) application to initiate clinical testing of TN-401, Tenaya Therapeutics. The evidence that this therapy might move to the clinic quickly is exciting for the field.

The reviewer appreciates the data on the efficacy of the therapy in iPSC-Cardiomyocytes: these data are important to demonstrate the applicability of the proposed therapy to human cardiac cells.

REVIEWERS' COMMENTS:

Reviewer #1 (Remarks to the Author):

The Authors present a preclinical study that tests a gene replacement strategy for treating arrhythmogenic cardiomyopathy caused by loss of function mutations in the cardiac Plakophilin gene (PKP2).

The investigation is performed in a PKP2 KO murine model. This model presents the limitation that, at variance with humans, the mice require the ablation of the protein to approximate the phenotypical traits seen in the heterozygous state in Humans. To address the limitation of the murine model, the authors also perform experiments in human induced pluripotent stem cell-derived cardiomyocytes.

The authors conclude that their studies demonstrate that “Restoration of PKP2 expression led to a highly coordinated and durable correction of structural genes encoding desmosome, sarcomere, and Ca²⁺-handling system, and corrections of multiple signaling pathways of metabolism, inflammation, and apoptosis.

Detailed comments to the authors

Lines 89-92 The authors state that (they) propose that cardiac AAV9:PKP2 could be a beneficial gene therapy approach to reduce ventricular arrhythmias, slow down adverse right ventricular remodeling, improve heart function and reduce mortality in ARVC patients with PKP2 mutations. This hypothesis will be tested in a First in Human clinical study as the authors state that

FDA has provided clearance of an Investigational New Drug (IND) application to initiate clinical testing of TN-401, Tenaya Therapeutics. The evidence that this therapy might move to the clinic quickly is exciting for the field.

The reviewer appreciates the data on the efficacy of the therapy in iPSC-Cardiomyocytes: these data are important to demonstrate the applicability of the proposed therapy to human cardiac cells.

We thank the reviewer for the comments.